# Using free air $CO_2$ enrichment data to constrain land surface model projections of the terrestrial carbon cycle

Nina Raoult[1*], Louis-Axel Edouard-Rambaut[2], Nicolas Vuichard[1], Vladislav Bastrikov[3], Anne Sofie Lansø[4], Bertrand Guenet[5], and Philippe Peylin[1]

[1]Laboratoire des Sciences du Climat et de l'Environnement, LSCE/IPSL, CEA-CNRS-UVSQ, Université Paris-Saclay, 91191, Gif-sur-Yvette, France
[2]CIRAD, UMR SELMET, Station de Ligne-Paradis – 7 chemin de l'IRAT – 97410 Saint-Pierre – Réunion, France
[3]Science Partners, Paris, France
[4]Department of Environmental Science, Aarhus University, Aarhus, Denmark
[5]Department of Geosciences Ecole normale supérieure (ENS), 24 Rue Lhomond, 75231 Paris Cedex 05, France
[*]now University of Exeter, United Kingdom

**Correspondence:** Nina Raoult (n.m.raoult2@exeter.ac.uk)

**Abstract.** Predicting the responses of terrestrial ecosystem carbon to future global change strongly relies on our ability to model accurately the underlying processes at a global scale. However, terrestrial biosphere models representing the carbon and nitrogen cycles and their interactions remain subject to large uncertainties, partly because of unknown or poorly constrained parameters. Parameter estimation is a powerful tool that can be used to optimise these parameters by confronting the model with

observations. In this paper, we identify sensitive model parameters from a recent version of the ORCHIDEE land surface model that includes the nitrogen cycle. These sensitive parameters include ones involved in parameterisations controlling the impact of the nitrogen cycle on the carbon cycle and, in particular, the limitation of photosynthesis due to leaf nitrogen availability. We optimise these ORCHIDEE parameters against carbon flux data collected on sites from the Fluxnet network. However, optimising against present-day observations does not automatically give us confidence in the future projections of the model,

given that environmental conditions are likely to shift compared to present-day. Manipulation experiments give us a unique look into how the ecosystem may respond to future environmental changes. One such type of manipulation experiment, the Free Air $CO_2$ Enrichment experiment (FACE), provides a unique opportunity to assess vegetation response to increasing $CO_2$ by providing data at ambient and elevated $CO_2$ conditions. Therefore, to better capture the ecosystem response to increased $CO_2$, we add the data from two FACE sites to our optimisations, in addition to the Fluxnet data. We use data from both $CO_2$

conditions of the Free Air $CO_2$ Enrichment experiment, which allows us to gain extra confidence in the model simulations using this set of parameters. We find that we are able to improve the magnitude of modelled productivity. Although we are unable to correct the interannual variability fully, we start to simulate possible progressive nitrogen limitation at one of the sites. Using an idealised simulation experiment based on increasing atmospheric $CO_2$ by 1% per year over 100 years, we find that optimising against only FluxNet data tends to imply a large fertilisation effect whereas optimising against FluxNet and

FACE data (with all nutrients limitation and acclimation of plant) decrease it significantly.

# 1 Introduction

Since the start of the industrial era, the atmospheric $CO_2$ concentration has risen from around 278 ppm in 1850 to 417.2 ppm in 2022 (Friedlingstein et al., 2022). Increases in atmospheric $CO_2$ lead to increases in leaf-scale photosynthesis and intrinsic water-use efficiency, which in turn have the potential to increase plant growth, vegetation biomass, and soil organic matter (Walker et al., 2021). Known as $CO_2$ sequestration, this process transfers carbon (C) from the atmosphere into terrestrial ecosystems. Indeed, terrestrial ecosystems currently remove about 30% of the $CO_2$ emitted by human activities each year (Friedlingstein et al., 2020). However, predicting how this carbon sink will evolve under increasing atmospheric $CO_2$ remains a challenge, especially due to the large uncertainties in the magnitude of carbon-climate feedbacks. Furthermore, the terrestrial ecosystem's ability to store carbon will be influenced by other processes, for example, nutrient limitations (Zaehle and Dalmonech, 2011) - most notably nitrogen (N), which is a key component controlling the carboxylation activity of the RubisCo in the photosynthetic tissue of the plant.

The large uncertainties in terrestrial carbon projections are largely related to the uncertainty in land surface models, including parametric uncertainty, which relates to the parameter values used in each parameterisation (Zaehle et al., 2005). The first land surface models were developed to provide a physical boundary to meteorology processes. As these models progressed, terrestrial biogeochemical cycles were implemented, simulating leaf gas exchange through Ball-Berry stomatal conductance and plant productivity based on Farquhar photosynthesis (Bonan, 2015). More recently, land surface models have moved from a big leaf model to multi-canopy schemes (Naudts et al., 2015), and started to include the nitrogen cycle and its constraints on the terrestrial carbon balance (e.g., LPJ: Prentice (2008), OCN: Zaehle and Friend (2010), ORCHIDEE-CN: Vuichard et al. (2019), CLM: Fisher et al. (2019)). However, with each new process and complexity added to the model, we add more internal model parameters, which in turn can add more uncertainty. Even though these parameters are generally chosen to represent measurable real-world quantities (e.g., leaf area, plant root depth), their default values are often issued from specific experiments studying the processes at different scales to those used in land surface models. Therefore it is important to confront simulated model outputs against independent data.

There are a lot of data with which we can evaluate model simulations from vast in situ networks (e.g., Fluxnet, Pastorello et al. (2020)) to state-of-the-art satellite retrievals (e.g., sentinel missions, Malenovskỳ et al. (2012)). It is important to evaluate land surface models against these types of data since they help increase confidence in the model simulations. Furthermore, these data can also be used to optimise models through parameter estimation. Parameter estimation methods can be used to perform parameter optimisation where uncertain parameters are tuned to minimise the difference between simulated model output and observed quantities. Fluxnet eddy-covariance data has already been used to optimise model parameters in most land surface models; e.g., ORCHIDEE (Kuppel et al., 2012), BETHY (Knorr and Kattge, 2005), JULES (Raoult et al., 2016), Noah (Chaney et al., 2016) and CLM (Post et al., 2017). However, evaluating and optimising against historical trends and present-day observations does not necessarily give us confidence in the future projections of the model, given that future environmental conditions are likely to shift compared to present-day (Wieder et al., 2019).

Fortunately, manipulation experiments give us a unique look into how the ecosystem may respond to future environmental change (Van Sundert et al., 2023). One such type of experiment, the Free Air $CO_2$ Enrichment experiment (FACE; Norby et al. (2010); Walker et al. (2018a)), provides a unique opportunity to assess vegetation response to increasing $CO_2$. FACE experiments are conducted across several vegetation types and typically consist of two types plots: one where $CO_2$ is fumigated to high concentrations and one left as a control.

In particular, two decade-long FACE experiments in temperate forests of the southeastern U.S. (Duke and Oak Ridge National Laboratory (ORNL)) have been predominately studied to test the representations of carbon–nitrogen cycle processes in land surface models. A full intercomparison of 11 land surface models (Medlyn et al., 2015) demonstrated how these data could be used to evaluate models looking at the effect of ambient and elevated $CO_2$ on water (De Kauwe et al., 2013), carbon (De Kauwe et al., 2014) and nitrogen (Walker et al., 2014; Zaehle et al., 2014). These two sites were further used in Wieder et al. (2019), where they showed how these experimental manipulations could be incorporated into the model benchmarking tools to help increase confidence in terrestrial carbon cycle projections. FACE experiments can also be used to identify processes that are not well caught by land surface models. For instance, Walker et al. (2019) showed that elevated $CO_2$ changed carbon allocation to the wood, and none of the models tested were able to reproduce this observation. Combined with warming experiments within a factorial design, the FACE experiments can also be very useful to evaluate how much the model are able to reproduce the single effect of elevated $CO_2$ versus the effect of elevated $CO_2$ when other drivers are changing (De Kauwe et al., 2017). More recently, Sulman et al. (2019) used these two sites to test the effect of adding symbiotic nutrient acquisition strategies to land surface models and Caldararu et al. (2020) to assess a whole-plant growth optimality approach in improving the representation of leaf nitrogen content compared to existing empirical approaches. The two sites are also the sites we focus on in this study.

We use these sites to check whether the parameterisations and parameters used in a land surface model are able to capture the ecosystem response to increased $CO_2$. Furthermore, by optimising a land surface model to both ambient and elevated conditions simultaneously, we gain extra confidence in the model projections using this single set of parameters. Although ideally we would want to calibrate under ambient conditions and test the model under elevated conditions, known model structural errors do not guarantee that the model is able to predict changes under different conditions. As such, we provide an alternative approach to model calibration, maximising the available information content of the optimisations. Our study is the first, to our knowledge, to do this with a global land surface model.

Using the ORCHIDEE land-surface model as an example, in this paper, we show the potential of using manipulation sites to not only optimise unknown model parameters but also increase confidence in the optimised model projections by reducing parameter uncertainty. Furthermore, by optimising parameters linked primarily to the nitrogen cycle, as well as considering nitrogen-limited FACE sites, we get an insight into the nitrogen-limiting effect on the fertilising effect of $CO_2$. This study also looks at how FACE data can complement Fluxnet data in a general optimisation procedure. As such, we aim to answer the following questions:

- Using parameter estimation, can we improve the representation of the simulated productivity of the new nitrogen-version of ORCHIDEE over Fluxnet and FACE sites (under both ambient and elevated conditions)?

– What is the benefit of adding FACE data on top of Fluxnet data when optimising a land surface model?

– How does the future evolution of terrestrial productivity change when simulated using different sets of optimised parameter values?

– Can these experiments help us to describe better the future fertilising effect of $CO_2$ under possible nitrogen limitation?

## 2 Methods

### 2.1 Model

#### 2.1.1 The ORCHIDEE land surface model

The ORCHIDEE (ORgainzing Carbon and Hydrology in Dynamic Ecosystems) model is a global terrestrial ecosystem model developed at IPSL (Institut Pierre Simon Laplace, France). It simulated the energy (Ducoudré et al., 1993), water (de Rosnay and Polcher, 1998), carbon (Krinner et al., 2005), and nitrogen (Zaehle and Friend, 2010; Vuichard et al., 2019) exchanges between the land surface and the atmosphere. This model can be run at various spatial resolutions, ranging from site to global 100 simulations, and over different timescales from one day to thousands of years. ORCHIDEE can be run as a stand-alone model driven by meteorological forcing or as part of the IPSL Earth System Model (Boucher et al., 2020; Lurton et al., 2020).

In ORCHIDEE, the different types of vegetation are discretised in Functional types (PFTs, Plant Functional Types) defined by plant metabolism, phenology, type of leaves and local climate. There are a total of 15 PFTs in ORCHIDEE; eight for the forests, four for the grasslands, two for the crops and one for bare soil. The model describes the different stocks of biomass in 105 the whole soil-plant continuum. There are nine stocks of biomass in the plant; the leaf, the above and below-ground sapwood, the above and below-ground heartwood, the fruits, the roots, and the long-term and short-term (available to use) reserves. For litter, there are six carbon stocks; metabolic, structural and woody above- and below-ground. Finally, there are four stocks for the soil organic matter; surface, active, slow and passive.

The litter pools are limited by the fall and death of tissues. The pools of organic matter in the soils are alimented by the 110 decomposition of the organic matter in the different pools of the litter. The decomposition of the organic matter is characterised by a fixed residence time for each litter and/or soil pool modulated by environmental conditions.

The carbon/nitrogen ratio of leaf biomass is variable, controlled by a supply/demand scheme while the C/N ratio of the other plant pools is a fixed proportion of the leaf C/N ratio. A specific C/N ratio is set for each soil pool, which varies as a function of the mineral nitrogen in soils. There are also additional mineral nitrogen pools in soils for ammonium, nitrate, nitrous oxides, 115 nitrogen oxides and dinitrogen. The inputs of nitrogen in the soil/plant system are considered deposition, fertiliser and manure inputs and biological fixation. Nitrogen losses are associated with leaching, lixiviation and emissions of ammonia, nitrous oxide, nitrogen oxides and dinitrogen.

The nitrogen component for ORCHIDEE was first developed and evaluated inside OCN, a version of the ORCHIDEE model (Zaehle and Friend, 2010). However, at the time, it was not embedded into the operational ORCHIDEE version used in

coupled experiments. This component has been recently updated and is now included in default ORCHIDEE runs (Vuichard et al., 2019). This has notably permitted studies of the interactions between the carbon and nitrogen cycles and their effect on gross primary production (GPP). The version of ORCHIDEE we use in our study (ORCHIDEEv3, r6863) is more recent than the one used (Vuichard et al., 2019, r4999). ORCHIDEEv3 (r6863) includes the latest developments of the main ORCHIDEE model (mainly small bug fixes). Furthermore, it includes updates to a few specific N-related processes, notably growth and maintenance respiration. Although this version has been used in the multi-model ensemble for the Global Carbon Budget 2020 (Friedlingstein et al., 2022), it has not yet been optimised against independent data. As such, the initial fit of the model to the Fluxnet data is different than that shown in Vuichard et al. (2019).

### 2.1.2 Model parameters

An initial list of parameters was compiled based on parameters used in past ORCHIDEE optimisations. This was extended to include parameters of the new nitrogen module selected using the expert knowledge of the module developers. Using a Morris sensitivity analysis (Morris, 1991), we remove all parameters to which the different modelled outputs tested (i.e., net primary product (NPP) and leaf-area index (LAI)) showed no sensitivity. All remaining parameters are optimised in this study (Table 1). These parameters represent key parameters of the model controlling photosynthesis, carbon and nitrogen allocation, respiration and global nitrogen cycle behaviour (full descriptions can be found in Appendix A). In addition, the KSoil parameter is used to control the initial carbon and nitrogen stocks. This parameter makes up for the fact that we cannot reconstruct each site's land-use history and its impacts on the present-day soil carbon stocks. Instead, we add the KSoil parameter in the optimisation, a multiplication factor applied on some soil carbon and nitrogen pools (slow, passive and labile) to change their initial values. A similar parameter has been used in many previous ORCHIDEE optimisation studies to control the initial carbon stocks of the model (e.g., Santaren et al., 2007; Kuppel et al., 2012; Bastrikov et al., 2018).

For each PFT, the Morris score for each parameter is normisalised by the most sensitive parameter. The normalised Morris sensitivity scores are shown in Table 1 and help us understand which are the most sensitive parameters. We see that for sites with a strong seasonal cycle, i.e., TeBS sites, the specific leaf area (SLA) phenology parameters are most sensitive. For the evergreen sites, two of the nitrogen parameters $NUE_{opt}$ and $K_{LAtoSA, max}$ gain importance ranking as highly as SLA.

### 2.2 Parameter estimation framework

We perform optimisations by relying on a Bayesian framework to include prior knowledge on the parameters ($\mathbf{x}_b$). Assuming that the errors associated with data observation, model output and parameters follow Gaussian distributions (Santaren et al., 2014), we seek to obtain a posterior optimal parameter set $\mathbf{x}_{opt}$ which corresponds to the minimum of the cost function $J(\mathbf{x})$:

$$J(\mathbf{x}) = (M(\mathbf{x}) - \mathbf{y})^T \mathbf{R}^{-1} (M(\mathbf{x}) - \mathbf{y}) + (\mathbf{x} - \mathbf{x}_b)^T \mathbf{B}^{-1} (\mathbf{x} - \mathbf{x}_b). \tag{1}$$

For a given parameter set $\mathbf{x}$, $J(\mathbf{x})$ measures the mismatch between observations $\mathbf{y}$ and the corresponding model outputs $M(\mathbf{x})$, and the mismatch between the prior, or background, parameter set $\mathbf{x}_b$ and $\mathbf{x}$. Each of these terms is weighted by their error covariances matrices, $\mathbf{R}$ and $\mathbf{B}$ for the observations and parameters respectively (Tarantola, 2005). In this study, we set both

**Table 1.** List of parameters used for the optimization with descriptions, default (prior) model values, ranges of variation, and normalised Morris scores denoting the relevant importance of each parameter (labelled "rk" for rank) - darker squares means more sensitive.

| Parameter | Description | Temperate Broadleaf Summergreen (TeBS) | | | | Temperate Needleleaf Evergreen (TeNE) | | | |
|---|---|---|---|---|---|---|---|---|---|
| | | min | prior | max | rk | min | prior | max | rk |
| Nitrogen-related processes | | | | | | | | | |
| $CTE_{bact}$ | Denitrification activity of bacteria (-) | 1e-05 | 3e-05 | 1e-04 | | 1e-05 | 3e-05 | 1e-04 | |
| $CN_{leaf, max}$ | Maximum C/N ratio of the leaves ($g_C[g_N]^{-1}$) | 36 | 45 | 54 | | 60 | 75 | 90 | |
| $CN_{leaf, min}$ | Minimum C/N ratio of the leaves ($g_C[g_N]^{-1}$) | 11 | 16 | 22 | | 18 | 28 | 38 | |
| $k_N$ | Extinction ratio of N through the canopy (-) | 0.13 | 0.15 | 0.18 | | 0.13 | 0.15 | 0.18 | |
| $FCN_{root}$ | N/C ratio of the roots/wood used to calculate allocation relative to | 0.6 | 0.82 | 1.2 | | 0.6 | 0.86 | 1.2 | |
| $FCN_{wood}$ | the leaf N/C ratio (-) | 0.06 | 0.087 | 0.12 | | 0.06 | 0.087 | 0.12 | |
| $NUE_{opt}$ | Nitrogen use efficiency of Vcmax ($\mu mol\ CO_2 s^{-1}\ [g_{Nleaf}]^{-1}$) | 23 | 33 | 43 | | 10 | 17 | 30 | |
| $R_{leaf}$ | Fraction of N leaf/root that is recycled when leaves are senescent (-) | 0.4 | 0.5 | 0.6 | | 0.4 | 0.5 | 0.6 | |
| $R_{root}$ | | 0.1 | 0.2 | 0.3 | | 0.1 | 0.2 | 0.3 | |
| z | Root profile (m) | 0.2 | 0.8 | 3 | | 0.25 | 1.0 | 4.0 | |
| $VMAX_{UPTAKE}$ | Maximal uptake capacity of roots for ammonium and nitrates (-) | 2 | 3 | 4 | | 2 | 3 | 4 | |
| Allocation | | | | | | | | | |
| $K_{LAtoSA, max}$ | Maximum leaf to sapwood area ratio ($m^2 gC^{-1}$) | 4000 | 9000 | 9900 | | 3000 | 5768 | 7500 | |
| $K_{LAtoSA, min}$ | Minimum leaf to sapwood area ratio ($m^2 gC^{-1}$) | 600 | 7200 | 9900 | | 450 | 4614 | 7500 | |
| $K_{root}$ | Fine root specific conductivity ($m^3 kg^{-1} s^{-1} MPa^{-1}$) | 3e-07 | 4e-07 | 5e-07 | | 1e-09 | 5e-09 | 1e-08 | |
| $K_{sap}$ | Maximal sapwood specific conductivity ($m^2 s^{-1} MPa^{-1}$) | 1e-04 | 3e-04 | 4e-04 | | 1e-05 | 3e-05 | 1e-04 | |
| Phenology | | | | | | | | | |
| SLA | Specific leaf area at the time of the leaf productions ($m^2 g^{-1}$) | 0.013 | 0.026 | 0.05 | | 0.009 | 0.004 | 0.02 | |
| $SLA_{init}$ | Initial Specific leaf area at the bottom of the canopy ($m^2 g^{-1}$) | 0.02 | 0.03 | 0.04 | | 0.034 | 0.044 | 0.054 | |
| $L_{agecrit}$ | Critical leaf age (days) | 90 | 180 | 240 | | 610 | 910 | 1210 | |
| $L_{fall}$ | Leaf fall (-) | 8 | 10 | 12 | | N/A | N/A | N/A | |
| $T_{senes}$ | Critical temperature for senescence (°C) | 10 | 16 | 22 | | N/A | N/A | N/A | |
| Photosynthesis (carbon assimilation) | | | | | | | | | |
| $k$ | Extinction ratio of the light through the canopy (-) | 0.3 | 0.5 | 1.0 | | 0.3 | 0.5 | 1.0 | |
| $A_1$ | Empirical factors involved in the calculation of fvpd (-, $kPa^{-1}$) | 0.7 | 0.85 | 0.9 | | 0.7 | 0.85 | 0.9 | |
| $B_1$ | | 0.1 | 0.14 | 0.18 | | 0.1 | 0.14 | 0.18 | |
| Respiration | | | | | | | | | |
| $FRAC_{growthresp}$ | Fraction of the GPP that is lost to growth respiration (-) | 0.2 | 0.28 | 0.36 | | 0.2 | 0.28 | 0.36 | |
| $Q_{10}$ | Parameter determining the temperature dependency of the heterotrophic respiration (-) | 0.0 | 0.69 | 1.1 | | 0.0 | 0.69 | 1.1 | |
| Spinup parameters (site dependent) | | | | | | | | | |
| KSoil | Multiplicative factor for initial soil carbon & nitrogen stocks (-) | 0.5 | 1 | 2 | | 0.5 | 1 | 2 | |

matrices to be diagonal. For $\mathbf{B}$, we define the prior distribution of each parameter to be 40% of the prior range. For $\mathbf{R}$, we define the observation error (variance) as the mean-squared difference between the observations and the prior model simulation so that this variance reflects not only the measurement errors but also the model errors. Furthermore, since we do not consider

error covariances, $\mathbf{R}$ is diagonal and therefore we can decompose the first term of Eq. 1 into different terms for each assimilated datastream:

$$J(\mathbf{x}) = k_{\text{Flx}}(M_{\text{Flx}}(\mathbf{x}) - \mathbf{y}_{\text{Flx}})^T(\sigma_{\text{Flx}}{}^{-1})(M_{\text{Flx}}(\mathbf{x}) - \mathbf{y}_{\text{Flx}}) + k_{\text{FACE}}(M_{\text{FACE}}(\mathbf{x}) - \mathbf{y}_{\text{FACE}})^T(\sigma_{\text{FACE}}{}^{-1})(M_{\text{FACE}}(\mathbf{x}) - \mathbf{y}_{\text{FACE}})$$
$$+ (\mathbf{x} - \mathbf{x}_b)^T\mathbf{B}^{-1}(\mathbf{x} - \mathbf{x}_b) \quad (2)$$

where Flx and FACE subscripts are used to denote the FLUXNET and FACE parts of the equation; $k_i$ denotes the weighting using for each datastream, $\sigma_i$ denotes the observational error, and $M_i$ and $\mathbf{y}_i$ denote modelled and observed data streams.

There exist many different approaches we can use to find the set of parameters which minimise $J(x)$. These range from simple manual tuning, which are very computationally demanding and inefficient, to more complex algorithms either based on deterministic gradient descent methods or stochastic random search methods. Using "ORCHIDAS", the ORCHIDEE data

assimilation tool developed at the Laboratoire des Sciences du Climat et de l'Environnement (Bastrikov et al. (2018)), we performed a couple of preliminary experiments to determine which algorithm to use. We tested a gradient descent method based on the L-BFGS-B algorithm (limited memory Broyden–Fletcher–Goldfarb–Shanno algorithm with bound constraints BFGS; Byrd et al. (1995)) and a random search method based on the genetic algorithm (GA; Goldberg and Holland (1988); Haupt and Haupt (2004)). We found that the GA method outperformed the gradient method in reducing the cost function.

These initial results are coherent with Bastrikov et al. (2018)'s study, which optimised the gross primary productivity (GPP) and latent heat fluxes of a former version of ORCHIDEE against a number Fluxnet site measurements and also found that the GA algorithm outperformed the other methods, notably by allowing a full exploration of all parameter space.

The genetic algorithm consists in applying the laws of evolution to our set of parameters by considering the set of parameters as a chromosome, with each parameter as a gene. At each iteration, the algorithm fills $k$ chromosomes with parameter values.

The first pool of chromosomes is created by randomly perturbing the value of the parameter. For the following iterations, the chromosomes are created from the previous iterations' chromosomes. Two processes come into play; a) a crossover process, where we have an exchange of genes between two chromosomes, and b) a mutation process, where random genes are perturbed. To ensure that the best chromosomes get the most descendants, each chromosome of each iteration gets ranked in function of the cost associated with the parameter's value in the chromosome.

## 2.3  In situ data

In this study, we consider two sites from the FACE network in nitrogen-limited temperate forest ecosystems; Oak-Ridge (ORNL; Norby et al., 2010) - a site dominated by broad-leaf deciduous forests (TeBS, for Temperate Broadleaf Summergreen Forests) and Duke (DUKE; McCarthy et al., 2010) - a site dominated by needle-leaf evergreen forests (TeNE, for Temperate Needleleaf Evergreen Forests). The data for these sites come from the FACE Model Data Synthesis project (Walker et al.,

2018a, b, https://facedata.ornl.gov/facemds/). For each site, we use the data from two experimental plots (with their associated

**Table 2.** List of in situ FACE and Fluxnet sites used in the study. The Fluxnet sites are labelled by a country code (first two letters) and site name (last three letters). The FACE sites are both found in the US. The period corresponds to the available years of data for each of the sites.

| Temperate Broadleaf Summergreen (TeBS) | | | Temperate Needleleaf Evergreen (TeNE) | | |
|---|---|---|---|---|---|
| Site id | Years | Coordinates | Site id | Years | Coordinates |
| **Free Air CO$_2$ Enrichment experiment sites** | | | | | |
| ORNL | 1999-2008 | 35.54, -84.20 | DUKE | 1996-2007 | 35.58, 70.5 |
| **FLUXNET2015 sites** | | | | | |
| DE-Hai | 2000-2012 | 51.08, 10.4 | CZ-Bk1 | 2004-2008 | 49.50, 18.54 |
| DK-Sor | 1996-2014 | 55.49, 11.64 | DE-Tha | 1996-2014 | 50.96, 13.57 |
| FR-Fon | 2005-2014 | 48.48, 2.78 | FR-LBr | 1996-2008 | 44.72, -0.77 |
| IT-Col | 1996-2014 | 41.85, 13.59 | IT-Lav | 2003-2014 | 45.96, 11.28 |
| IT-PT1 | 2002-2004 | 45.20, 9.06 | IT-Ren | 1998-2013 | 46.57, 11.43 |
| IT-Ro1 | 2000-2008 | 42.41, 11.93 | IT-SRo | 1999-2012 | 43.73, 10.28 |
| IT-Ro2 | 2002-2012 | 42.39, 11.92 | NL-Loo | 1996-2013 | 52.17, 5.74 |
| US-Ha1 | 1991-2012 | 42.54, -72.17 | RU-Fyo | 1998-2014 | 56.46, 32.92 |
| US-MMS | 1999-2014 | 39.32, -86.41 | US-Blo | 1997-2007 | 38.90, -120.63 |
| US-UMB | 2000-2014 | 45.56, -84.71 | US-GLE | 2004-2014 | 41.37, -106.24 |
| US-WCr | 1999-2014 | 45.81, -90.08 | US-Wi4 | 2002-2005 | 46.74, -91.17 |

error bars); one with unaffected atmospheric CO$_2$, i.e. ambient (AMB), and one with elevated atmospheric CO$_2$ (ELE). Although the DUKE experiment also has ammonium nitrate treatments at half of its plots from 2005 onwards (Feng et al., 2010), we only consider the data from the plots without nitrogen fertilization.

The version of ORCHIDEE we use in this study has yet to be optimised against Fluxnet data using a Bayesian framework, as it was done with previous nitrogen-free versions of the model (e.g., Kuppel et al., 2012; Peylin et al., 2016). Therefore, we also consider TeBS and TeNE sites from the FLUXNET2015 dataset (Pastorello et al., 2020). This dataset provides gap-filled half-hourly meteorological data measured at each site (air temperature, humidity, pressure, wind speed, rainfall and snowfall rates, shortwave and longwave incoming radiation; see Vuichard and Papale (2015)). It also provides net carbon flux measurements, as such net ecosystems exchange (NEE) further split into gross primary production (GPP) and total ecosystem respiration (TER) following a classical night-time vs day-time flux partition Lasslop et al. (2010). For each of the two vegetation types, sites with over 60% vegetation coverage are kept. We exclude sites with too large discrepancies with the prior model output, such as with no apparent seasonal cycle, large data gaps, or with only one year of data. The list of in situ sites used can be seen in Table 2 partitioned by vegetation type.

## 2.4 Performed experiments

Before performing the optimisations, for each of the sites in this study, a two-step spin-up is performed. The first step helps to put the prognostic variables, including vegetation state, soil carbon pools, and soil moisture content at equilibrium. The available meteorological forcing is cycled over several millennia (with pre-industrial $CO_2$ concentrations) to ensure that the long-term net carbon flux was close to zero. After reaching the equilibrium, a second simulation is performed (transient) from the year 1860 to one year before the first forcing year while increasing $CO_2$ concentration at each simulation year following global historical observations.

Before performing the optimisations, we also conduct a sensitivity analysis on the parameters (as described in Sect. 2.1.2 and shown in Table 1). A sensitivity analysis tests how different the model outputs change with respect to different parameters. This is done to ensure that only parameters showing some sensitivity to the model outputs are used in the optimisation and therefore minimising the risk of using parameters that are weakly constrained by the fluxes. This is an important step since we want to avoid constraining parameters that will have a small impact on the optimisation but have the potential to significantly degrade the model-data fit against processes not included in the calibration.

Once spun up and with the list sensitive parameters, we perform two main sets of optimisations always starting from this spinup. The first is over the Fluxnet sites only while the second also includes data from the FACE sites. Due to the $CO_2$ fumigation over FACE sites, NEE is not measured at these sites, and therefore, GPP and TER estimates cannot be derived. Instead, for the FACE sites, we have annual net primary production (NPP) and daily leaf area index (LAI) data. Throughout this study, we perform multi-site (MS) optimisations, i.e., optimizations executed over multiple sites of the same PFT simultaneously in order to find one common set of optimised parameters. Each optimization is run for 20 iterations, which we found to be sufficient for the system to converge. For each iteration, 32 chromosomes are used i.e., 32 different combinations of parameter values. We leave the last year of each Fluxnet site out of the optimisation to have independent data for the validation step of the analysis.

The first set of optimizations test two different combinations of gross and net carbon fluxes:

- **Flx$_{GR}$:** two MS optimizations against daily GPP and TER, one for all the TeBS sites and one for all the TeNE sites.

- **Flx$_{GN}$:** two MS optimizations against daily GPP and NEE, one for all the TeBS sites and one for all the TeNE sites.

In each case, two fluxes are used in optimisations. Note that GPP and TER are derived from NEE with NEE=TER-GPP. This means that these data are model-derived estimates, which could introduce additional uncertainty to the results. However, by separating the fluxes we get a better understanding of the underlying mechanisms constraining two different ecosystem functions and are able better to diagnose the overestimations or underestimations of the assimilated processes, as initially discussed in Santaren et al. (2007). We are especially interested in the GPP constraint since this will give us an insight into plant productivity and will allow us to assess the $CO_2$ fertilising effect under nitrogen limitation. GPP is also directly used in the calculation of water use efficiency (WUE), here defined as the ratio between GPP and transpiration, one of the diagnostics we consider at the end of the study.

We further acknowledge that the data streams are not independent from each other. This poses a challenge when working in a Bayesian framework, especially when defining the $\mathbf{R}$ matrix in Eq. 1. Although there are methods for including the correlation between different data streams in the $\mathbf{R}$, these are relatively new and require a lot of extra analysis beyond the scope of this study. Instead, we rely on the standard method of inflating variances (Chevallier, 2007).

The optimal parameters found by optimising against the Fluxnet sites improve the fit to contemporary data. However, it is unclear whether the predicative skill of the model is improved. Therefore, after assessing the Fluxnet results, the next step is to incorporate the FACE sites. Using a simultaneous approach, the FACE and Fluxnet sites are optimised together in this second set of experiments. This approach ensures that the information is not lost between steps, as could be the case in step-wise approach when the optimisations are done one after the other. The optimisations are set up to give a higher weight to the single FACE site in each case, so that $k_{Flx}=1$ and $k_{FACE}=n$ in Eq. 2 where $n$ is the number of Fluxnet sites for the given PFT. Based on our results (see Sect. 3.1) and our motivation to better capture the productivity of the different ecosystems, we choose to focus on the former Fluxnet optimisation, i.e. the one against GPP and NEE. Each of the following FACE site experiments are performed simultaneously with a Flx$_{GN}$ optimisation over the relevant PFT:

– **Flx$_{GN}$-AMB:** two optimisations against annual NPP and daily LAI, one each for the DUKE and ORNL sites at ambient $CO_2$ concentrations, perform simultaneously with a GPP-NEE multisite Fluxnet optimisation.

– **Flx$_{GN}$-ELE:** two optimisations against annual NPP and daily LAI, one each for the DUKE and ORNL sites at elevated $CO_2$ concentrations, perform simultaneously with a GPP-NEE multisite Fluxnet optimisation.

– **Flx$_{GN}$-BOTH:** two optimisations against annual NPP and daily LAI, one each for the DUKE and ORNL sites with both ambient and elevated $CO_2$ concentrations simultaneously, perform simultaneously with a GPP-NEE multisite Fluxnet optimisation.

For the final part of this study, we consider the sensitivity of the simulated GPP, NPP, and WUE to $CO_2$ increase whilst keeping the other drivers constant. Each of the Fluxnet sites is tested by running idealised 100-year-long simulations starting from present-day atmospheric $CO_2$, 380 ppm, and increasing $CO_2$ by 1% per year, leading to a near tripling of $CO_2$ by the end of the simulation. This is done for both the prior and optimised model, using default model parameters and Flx$_{GN}$-BOTH model parameters, respectively.

## 3  Results and Discussion

### 3.1  Fluxnet optimisations

The nitrogen cycle version of the ORCHIDEE model used in this study has not yet been optimised against Fluxnet data, although it has been extensively tuned manually. Therefore, the first step is to see whether the fluxes over the TeBS and TeNE sites are well represented in the model and whether they can be optimised using observed data and the sensitive parameters identified in the Morris experiment (Table 1). For the Fluxnet optimisation, two tests are conducted. The best-performing

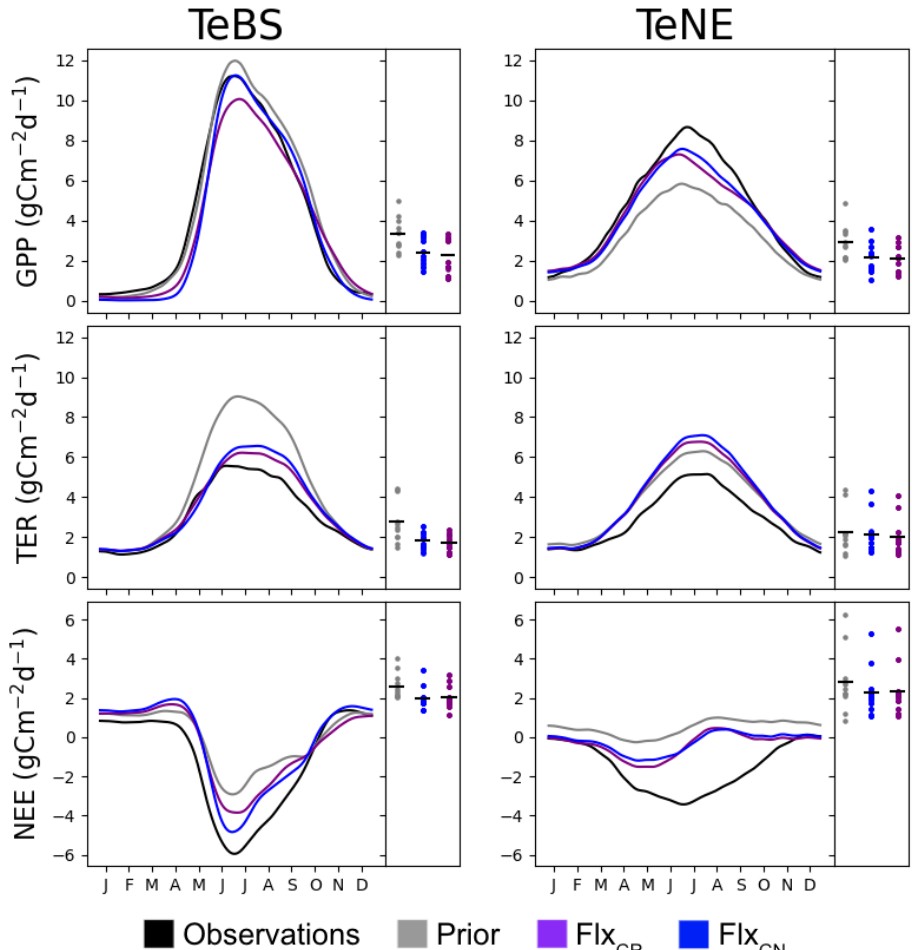

**Figure 1.** The main panel in each column shows the PFT-averaged mean seasonal cycles of daily observed and simulated GPP, TER, and NEE fluxes using different parameter values. The side panels show the model-data RMSD for the daily time series at each site, with the black horizontal bars showing the mean value across the sites.

optimisation will serve as the starting point for the optimisations including data from the FACE sites. In this section, we present the results from both MS Fluxnet optimisations: $Flx_{GR}$ and $Flx_{GN}$. Figure 1 shows the mean seasonal cycle across all Fluxnet sites for each PFT considered. We show the modelled GPP, TER and NEE fluxes against the observed time series.

For the deciduous sites (TeBS; Fig. 1 left-hand column), we see that both GPP and TER are overestimated by the prior model, and the NEE sink in underestimated. This overestimation is the most severe for TER, where the prior model simulates a maximum of approximately 9 $gCm^{-2}d^{-1}$ when the maximum TER observed is half that. In contrast, the overestimation for GPP is very slight found at and after the peak. Both optimisations improve the model-data fit against GPP by correcting the overestimation found after the peak. $Flx_{GN}$ performs the best, with the average seasonal peak now being the same as the

observations. Flx$_{GR}$ on the other-hand reduces peak below that observed. Similarly, both optimisation now starts production later in the year degrading the fit to the observations in the early months. In ORCHIDEE, deciduous sites lose all their leaves in winter and therefore, no photosynthesis occurs before the leaves start growing back in spring. In contrast, the observations never go to zero, implying there is undergrowth or evergreen vegetation present that we are not accounting for in the model set-up. When looking at the RMSD of the individual sites, we also see that Flx$_{GN}$ reduces the spread relatively to the prior.

For TER, both optimisations improve the model-data fit over the whole period with Flx$_{GR}$ performing slightly better. This is not surprising since this optimisation directly considers the TER component of NEE. The optimisations mainly change the magnitude of the peak and do not correct for its late timing. When looking at the RMSD, we can see that a group of sites are driving the overestimation of TER with values close to 4.5 - this is corrected for in both optimisations. For NEE, the Flx$_{GN}$ optimisation performs better than Flx$_{GR}$, especially in fitting the autumn month. However, because of the overestimation of

summer TER with this parameter set means we do not attain the minimum of the NEE trough. We note that for NEE, the posterior spreads of RMSE over all sites are the same for both optimisations.

For the evergreen sites (TeNE; Fig. 1 right-hand column), the optimisations improve the GPP underestimation in the prior model by increasing the peak by approximately 2.5 gCm$^{-2}$d$^{-1}$. H0wever, this falls short of correcting of the full overestimation, which is closer to 4 gCm$^{-2}$d$^{-1}$. The Flx$_{GN}$ optimisation performs best with the timing of the average peak closest to the

285 observed value. However, both optimisations degrade the average fit to TER, increasing the overestimation found when using the prior set of parameters. Both optimisations move the summer peak up by between 0.5-1 gCm$^{-2}$d$^{-1}$, with Flx$_{GN}$ increasing the most. When considering the fit of the individual sites (right-hard part of each panel), we note that two anomalous sites are driving this behaviour. These sites (IT-Lav and US-Wi4) have respiration rates much lower than the other sites. Since we cannot match the respiration rates, we cannot capture the full NEE dip in summer. However, the improved GPP with both

optimisations does mean that the NEE is slight improved compared to the prior mode.

In these optimisations, we have included a parameter, KSoil, which acts as a multiplicative factor of the initial soil and nitrogen pools (slow, passive and labile). We have used such a parameter in past ORCHIDEE optimisations at Fluxnet sites (e.g., Kuppel et al., 2012; Peylin et al., 2016; Bastrikov et al., 2018; Bacour et al., 2022) and found it played a large role in improving the model-data fit against respiration. Therefore, it seems counter-intuitive that we do not improve the fit to

295 respiration as much as expected when including KSoil in the ORCHIDEE v3 optimisations, especially for the TeNE sites. The past ORCHIDEE experiments all used a previous version of ORCHIDEE without the nitrogen cycle, so this factor solely acted on the carbon pools and heterotrophic respiration. Here KSoil multiplies both carbon and nitrogen pools to maintain the carbon/nitrogen ratio. However, by acting on the nitrogen pools, we directly impact on the mineralization rate and thus indirectly on plant N uptake, leaf N content, Vmax and, therefore, GPP. Whereas KSoil used only to impact soil respiration, it

now impacts both respiration and GPP, and so the optimisation needs to find a compromise to fit both. To adjust the respiration, Ksoil is decreased, reducing the carbon and nitrogen pools in the soils, but, at the same time, GPP is significantly reduced, deteriorating its fit to observations.

Overall, the ORCHIDEE model reasonably represents the TeBS and TeNE carbon fluxes, although respiration in the TeNE sites is high, even after optimisation. The Flx$_{GN}$ optimisation results in the best-simulated production for both types of vegetation.

## 3.2 Incorporating data from the FACE sites

Given the results from the previous section, and our motivation to improve the model performance regarding ecosystem productivity, we will further include the FACE data to the Flx$_{GN}$ optimisation.

### 3.2.1 Improving simulated NPP values

At ORNL, we can see in Fig. 2 that the uncalibrated version of ORCHIDEE (prior) overestimates the yearly NPP both under ambient and elevated conditions. This is consistent with prior GPP overestimation observed at the TeBS Fluxnet sites (Fig. 1). When using parameters from the Fluxnet only optimisation (Flx$_{GN}$), we partly reduce this overestimation. For both $CO_2$ conditions, including FACE data as an additional constraint to the optimisation (Flx$_{GN}$-AMB, Flx$_{GN}$-ELE and Flx$_{GN}$-BOTH) improves the estimation of NPP compared to solely relying on the data from the Fluxnet sites. Under all atmospheric $CO_2$ conditions, Flx$_{GN}$-ELE reduces the RMSD the most followed by Flx$_{GN}$-BOTH and then by Flx$_{GN}$-AMB. We would expect the latter to perform best at fitting NPP$_{AMB}$ since it uses the observations in the optimisation, unlike Flx$_{GN}$-ELE. However, as we will see later in Fig. 3, this is because the fit to LAI, the other part of the cost function, is improved most with the Flx$_{GN}$-AMB optimisation.

When considering the NPP ratio (elevated over ambient values) at ORNL, we see that the observations show a decreasing trend suggesting a possible progressive nitrogen limitation. The prior model is unable to capture this trend with a fixed ratio of around 1.3. Similarly, the Fluxnet only and the optimisation under ambient conditions do not mimic this decreasing trend. When using the FACE data from the elevated experiments in optimisation (Flx$_{GN}$-ELE and Flx$_{GN}$-BOTH, the resulting ratio trend is negative. The observations of NPP$_{AMB}$ and NPP$_{ELE}$ clearly show a slight decrease of NPP for the last years of the observations. Based on additional experiments, it has been shown that this limitation is due to nitrogen limitation (Norby et al., 2010). Again the prior does not capture the NPP decrease. For optimisations, Flx$_{GN}$-ELE and Flx$_{GN}$-BOTH may show a small signal, although it's not so obvious. This leads us to think that ORCHIDEE does not reproduce the dynamic of nitrogen in the soil well, notably the reduction of nitrogen availability for the plant, at least for this site. It is also possible that the model starts with too large a nitrogen content in the soil. Since parameter optimisation is insufficient at capturing this trend, we would instead need structural changes to the ORCHIDEE land surface model or possibly set a better initial nitrogen available content.

For DUKE, the prior model underestimates the NPP values under both $CO_2$ conditions, which again is coherent with GPP fit seen at the other TeNE sites (Fig. 1). The optimisation against Fluxnet data corrects this underestimation slightly. However, it is only by including the FACE data to the optimisation that we get the right magnitude and start getting the right inter-annual pattern. In each case, the optimisation performing best is the one that includes the relevant observations, i.e., Flx$_{GN}$-AMB under ambient conditions and Flx$_{GN}$-ELE under elevated conditions. The Flx$_{GN}$-BOTH optimisation is able to fit both the ambient and elevated data reducing the RMSD to a similar extent as the best optimisation. When considering the elevated over

ambient NPP ratio for DUKE, we see that the prior model and the Fluxnet-only optimisation ($Flx_{GN}$) perform the worst with some values below one. Values below one suggest the forest is less productive under increased atmospheric $CO_2$. Looking more closely at the ORCHIDEE simulations for these years and parameter settings, we found that the mean-annual GPP under elevated conditions is higher than GPP under ambient conditions. Therefore, changes autotrophic respiration are responsible

for the negative NPP ratio at the site. Maintenance respiration is a function of leaf nitrogen content; the more nitrogen, the more respiration. It is, therefore, possible that the maintenance respiration sensitivity to leaf nitrogen is too high in the model for TeNE under these parameter settings, at least at the DUKE site. The optimisations using FACE data do much better at simulating positive ratios with $Flx_{GN}$-BOTH performing best. However, none of the optimisations constantly achieve the high magnitude of around 1.3 in the observations.

### 3.2.2  Improving the fit of leaf-area index

During the FACE experiments, we also optimised the model against daily LAI values (Fig. 3) since this allows us to capture some of the sub-annual variability driving NPP. For ORNL, under ambient and elevated $CO_2$, the prior model overestimates the LAI, peaking late and keeping its leaves late into the winter. The optimisation against the Fluxnet sites partly fixes both of these issues, bringing forward the leaf fall and decreasing the seasonal peak to 65% of the observed peak. When adding the FACE data

under ambient conditions to the optimisation (i.e., $Flx_{GN}$-AMB) we get the best fit, although the peak is still underestimated. The optimisation against both atmospheric conditions ($Flx_{GN}$-BOTH) performs similarly to $Flx_{GN}$-AMB. When we only add the elevated data to the optimisations, we do the worst of the FACE optimisations, with the summer peak nearly half of the observed. This explains why $Flx_{GN}$-ELE outperformed $Flx_{GN}$-AMB in fitting NPP at ORNL (Fig. 2). By improving NPP to a large extent than $Flx_{GN}$-AMB, $Flx_{GN}$-ELE was unable to improve the LAI as much. This highlights one of the common issues

with multi-stream parameter estimation - improving the fit against one data stream can be part of a trade-off against another. Sometimes this can even be a degradation compared to the prior. Finally, although leaves are kept into the winter, unlike in the observations, the LAI seasonality is much improved in the optimisations compared to the prior model.

### 3.2.3  Posterior parameter values

In Fig. 4, we consider how the different parameters have changed by comparing results from the Fluxnet-only optimisation

and the optimisation including the FACE data. For the temperate broadleaf summergreen parameters it is hard to distinguish significant patterns. The posterior values sometimes move in the same direction for both optimisations (e.g., $K_N$, $FCN_{wood}$, $R_{root}$ and z), other times the parameters are pulled in different directions (e.g., $CN_{leaf,max}$ and $R_{leaf}$). When considering the most sensitive parameters, both optimisations agree with the direction of change, with the exception of NUEopt. For the photosynthetic parameters $A_1$ and $B_1$, we see that these are changed during the Fluxnet-only optimisation but are less important

in correcting the fit when the FACE data is included in the optimisation. Overall this highlights that adding the FACE data can pull the parameters in a different direction than when using only Fluxnet data. This indicates that there are likely compensating effects (or different repartitioning of model error). It is possible that with more data sets, the results would be more robust but this would need to be confirmed with the use and assimilation of additional data streams.

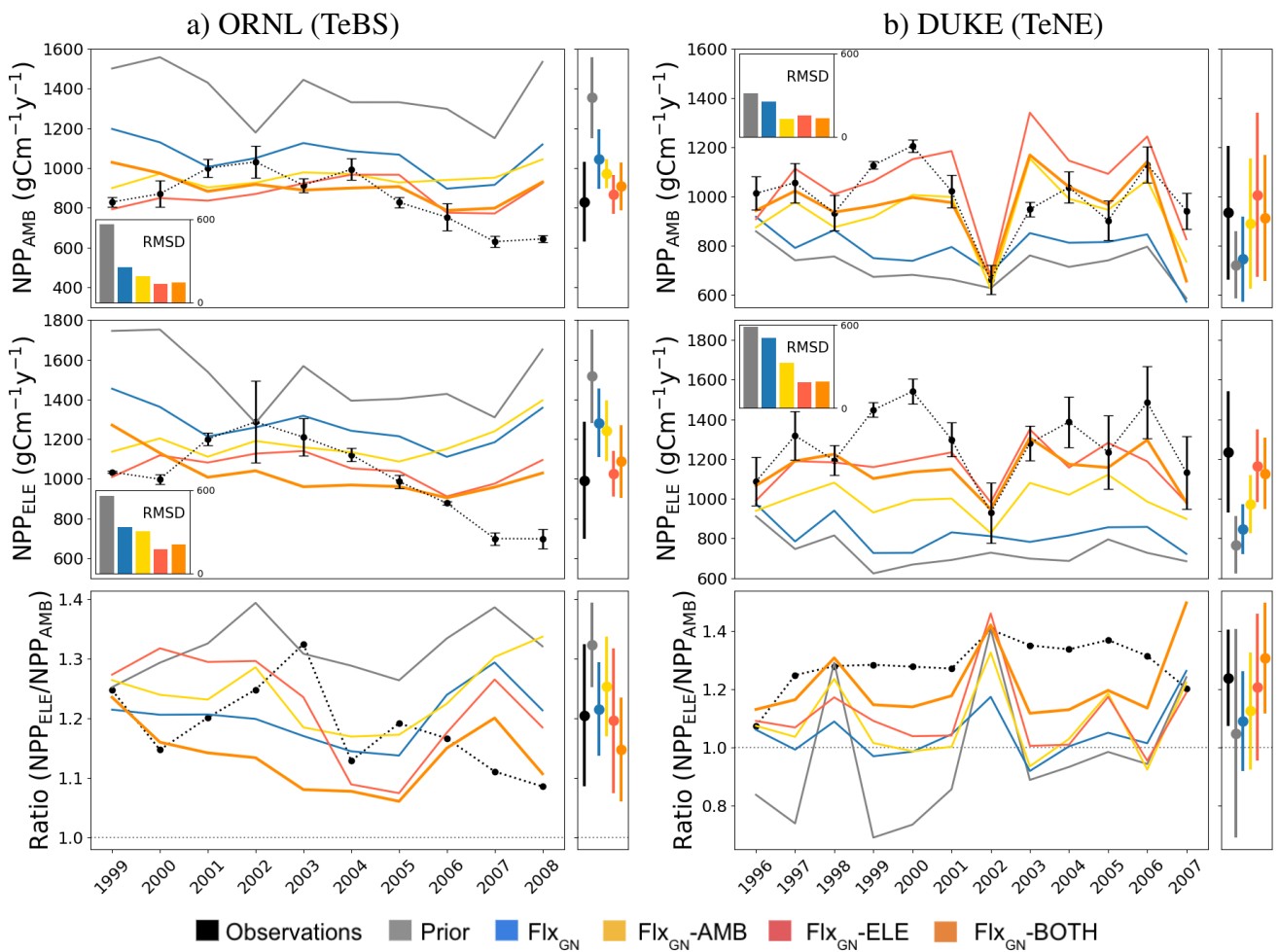

**Figure 2.** Time series of annual NPP under ambient (NPP_AMB; top) and elevated (NPP_ELE; middle) $CO_2$ conditions for each of the ORNL (a) and Duke (b) FACE sites. The ratios between NPP under elevated conditions and NPP under ambient conditions are shown in the bottom row, with a dotted-grey line indicating where the ratio in 1. The observations are shown in black, with error bars. The coloured lines represent different ORCHIDEE model simulations under different parameter sets - prior refers to the standard ORCHIDEE run using uncalibrated parameters, and the different "Flx" runs denote the data streams used in the optimisations (defined in Sect. 2.4). The bar chart insert in each time series panel shows the RMSD between the observations and each model run. The multi-annual NPP mean ± its standard deviation is shown next to each panel.

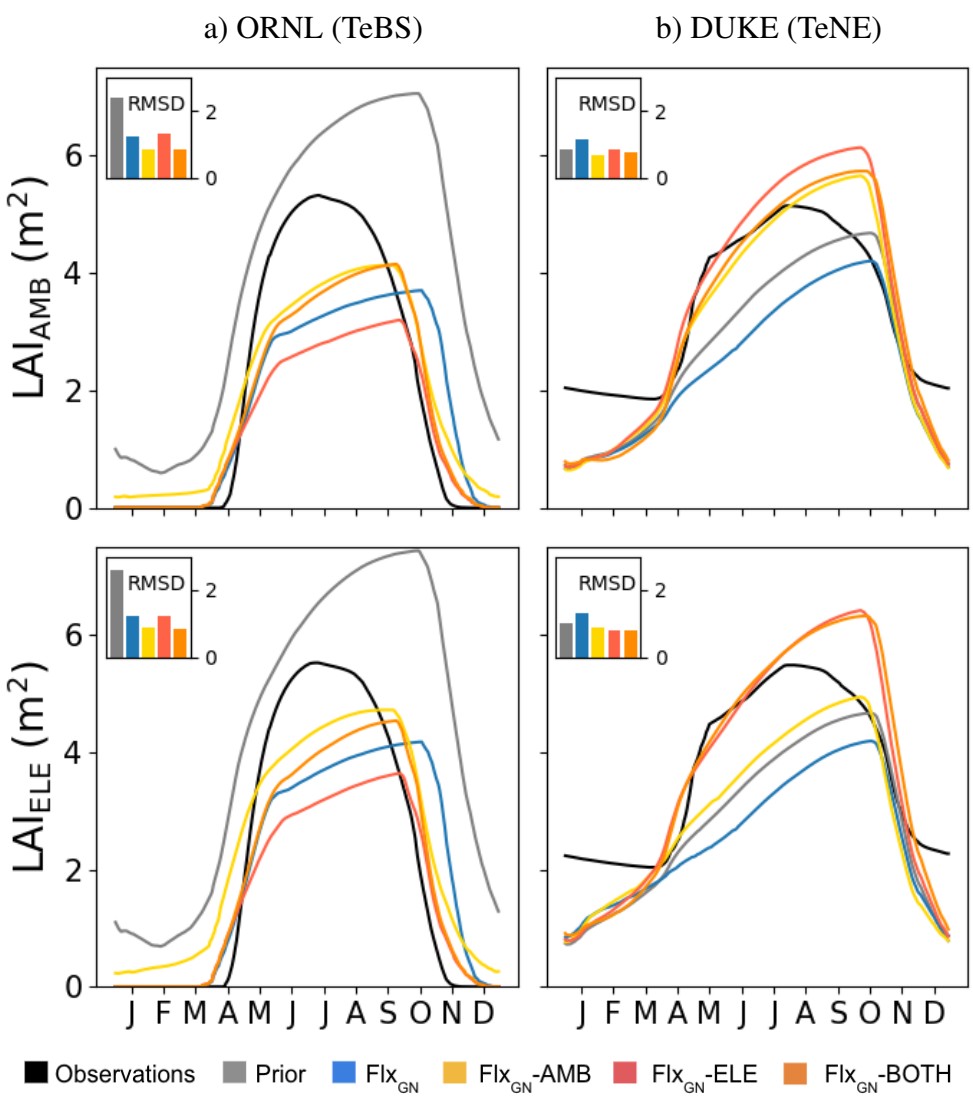

**Figure 3.** Model fit against seasonal LAI under different parameter sets at ORNl and DUKE. LAI under ambient $CO_2$ conditions ($LAI_{AMB}$) shown in the top row and elevated $CO_2$ conditions ($LAI_{ELE}$) on the bottom row. The bar chart inset in each panel shows the RMSD between the observations and the model.

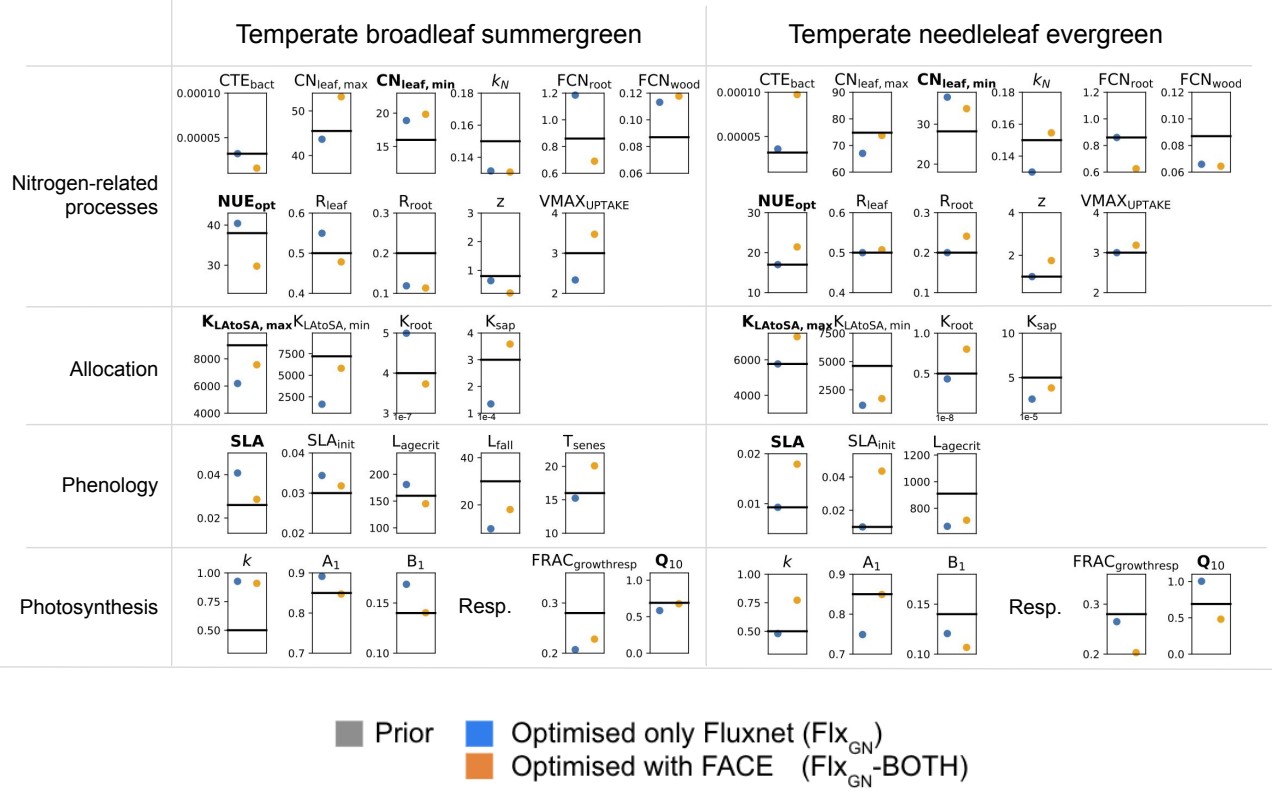

**Figure 4.** Posterior parameter values of the Fluxnet-only optimisation (blue, Flx_GN) and the optimisation incorporating FACE data under both atmospheric regimes (orange, Flx_GN-BOTH). The horizontal line represents the prior value of each parameter and each box spans the range of variation allowed for each parameter during the optimisation. Parameters highlighted in bold correspond to the parameters identified as most sensitive in the preliminary Morris experiment (see Table 1).

In contrast, for the temperate needleleaf parameters, nearly half (11/23) of the parameters were unchanged (or only slightly) during the Fluxnet-only optimisation. However, when optimised with the additional FACE data, these parameters changed greatly. These parameters include a number of the most sensitive parameters suggesting these are especially important in capturing the model response under both atmospheric regimes for needleleaf sites.

Although looking at these parameter values can be very informative - we must remember that there are complex interactions between the parameters and processes that will not be evident by just looking at these values.

### 3.2.4  Maintaining the fit to the Fluxnet sites

The misfit part of the cost function (i.e., measuring the difference between the model and observations) is made up of two components during the FACE optimisations. The first is the fit of the FACE site to annual NPP (Fig. 2) and daily LAI (Fig. 3)

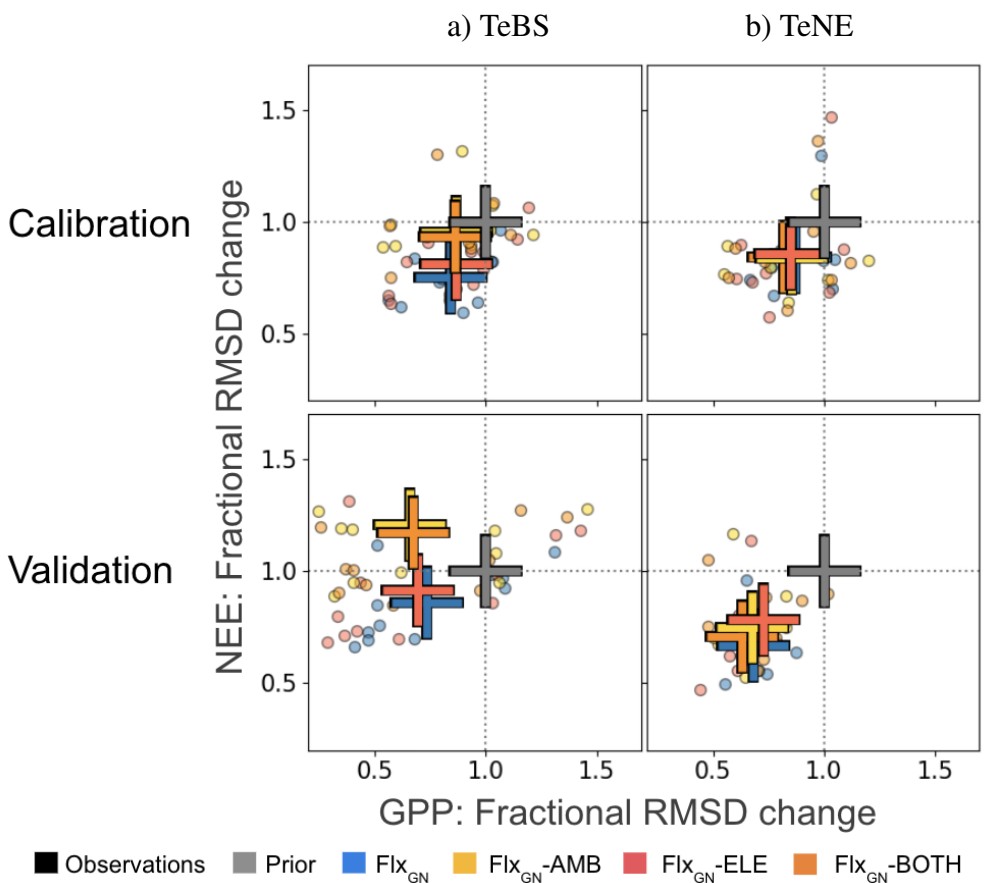

**Figure 5.** Fractional change in model-data RMSD for daily NEE and GPP at each site grouped by PFT obtained by the different optimisations. Fractional change is calculated by dividing the posterior RMSD by the prior RMSD (i.e., RMSD[$\mathbf{y}, M(\mathbf{x}_{opt})$]/RMSD[$\mathbf{y}, M(\mathbf{x}_b)$] using the notation in Sect. 2.2). Values below 1 represent an improvement in model-data fit, whereas values above 1 represent an increase in RMSD compared to the prior run. The crosses show the mean value across the sites for a given optimal parameter set. This figure complements Fig. 1, here showing the optimisations using the FACE data fit the timeseries compared to the prior model and the optimisation (GN) without FACE.

under ambient or evaluated conditions. The second is the fit of the multiple Fluxnet sites to daily GPP and NEE. Although the FACE component was weighted during the optimisation (see section 2.4), it is important that we maintain a good model-data fit for the Fluxnet sites to ensure confidence in the optimised parameter values. In Fig. 5, we consider the fractional change in model-data RMSD (calculated by dividing the posterior RMSD by the prior RMSD). In other words, this helps us see to what extent the calibration with FACE data changes the estimated fluxes shown in Fig. 1. For both vegetation types, the best improvement against both GPP and NEE is found when using parameters from the experiment solely optimising over the Fluxnet sites (Flx$_{GN}$). This is true for both the optimisation years, i.e., the years used in the optimisation, and the validation year, i.e., the year of independent data omitted from the optimisation. Adding further constraints by including FACE data reduces the improvement in model-data fit. After all, even though they are closely related, the FACE optimisations looked at improving the fit to NPP and LAI, not GPP and NEE. However, the sets of parameters obtained from these latter optimisations will give a better compromise for both the FACE and Fluxnet sites.

For the TeBS sites, on average, we improve the fit to GPP over the optimisation years to the same extent for all optimisations. Slightly higher reductions in fractional GPP RMSD change are observed for the validation year. However, for NEE, fractions in RMSD are different depending on the optimisation - with the ambient and both optimisations resulting in the smallest decreases for the calibration years. For the validation, these two optimisations degraded the fit for the validation year (20%).

In contrast, for the TeNE sites against the calibration years, the optimisations all give similar reductions in the model-data fit RMSD for both NEE and GPP. When confronted with the validation year, the response is more spread out, with the Fluxnet only optimisation performing the best, closely followed by Flx$_{GN}$-BOTH.

For the validation, we only used one year of data, which helps to explain why the results are more spread out. With one year of data, we are more susceptible to specifics of that given year instead of to trends over a longer period. Ideally, we would want to validate over a larger period. However, some of the sites in this study only had a few years in their data record, making this not possible.

## 3.3 Projections using the optimised models

To conclude this study, we test how the optimised model parameters impact the model responses to $CO_2$ increase. We especially want to consider the additional information gained from including FACE data in the optimisation. As such, for this last experiment, we use parameters of the Flx$_{GN}$ (Fluxnet only) and Flx$_{GN}$-BOTH (Fluxnet and FACE data) optimisations. The Flx$_{GN}$-BOTH optimisation resulted in the best compromise in simulating NPP at ambient and elevated conditions, as well as their ratio, while also maintaining a satisfying model-data fit to GPP over the Fluxnet sites.

To test the model response to $CO_2$ increase, we run an idealised 100-year-long experiment increasing $CO_2$ 1% per year over all the Fluxnet sites. This is similar to the experiment done in Vuichard et al. (2019) where the model was also assessed with GPP, transpiration and WUE. Note that the prior projections shown here are similar to those obtained in Vuichard et al. (2019). We first note that the optimised models predict lower starting values of GPP, Transpiration and WUE for TeBS when compared to the prior model (Table 3). This is consistent with Fig. 1 and 2, where for TeBS, the prior had the highest productivity. Similarly, the behaviour for TeNE sites mirrors earlier findings - the prior model underestimated productivity. The optimisations

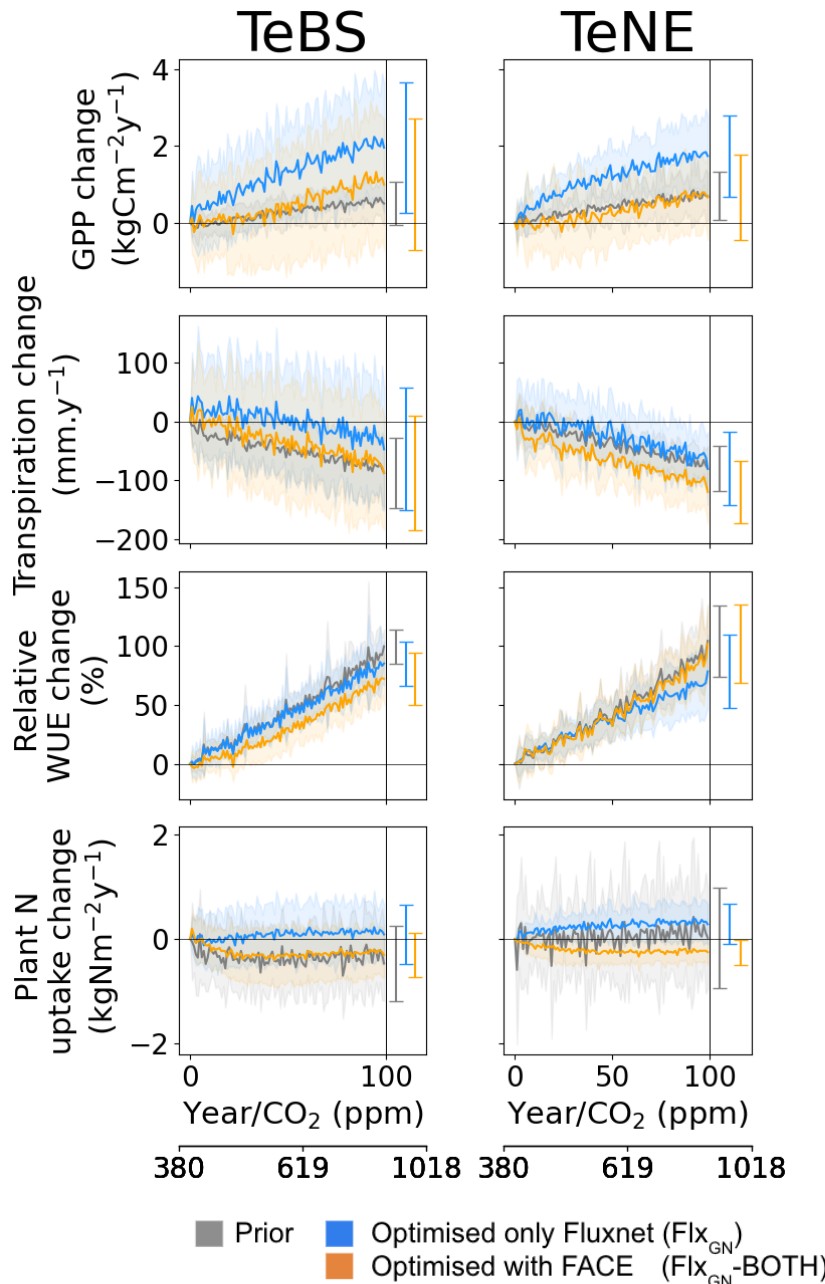

**Figure 6.** Effect of changes in the atmospheric $CO_2$ concentration on GPP, NPP, Water Use Efficiency (WUE) and Plant N uptake for different model parameter sets. Atmospheric changes are represented by 1%$CO_2$ increase per year. The prior model (grey) and optimised model (Flx$_{GN}$-BOTH; orange) were run at each Fluxnet site (see Table. 2). The thick lines represent the mean simulated across all sites, while the shaded areas represent the standard deviation. The mean and standard deviation at the end of the 100-year simulation are shown in the right-hand side of each panel.

| | TeBS | | | TeNE | | |
|---|---|---|---|---|---|---|
| | Prior | Optimised | | Prior | Optimised | |
| | | Fluxnet only | Fluxnet & FACE | | Fluxnet only | Fluxnet & FACE |
| GPP ($kgCm^{-2}y^{-1}$) | 5.22 + 0.55 | 3.58 + 1.95 | 4.03 + 0.99 | 2.85 + 0.73 | 3.93 + 1.73 | 4.59 + 0.66 |
| Transpiration ($mm.y^{-1}$) | 417.29 - 83.43 | 342.93 - 47.06 | 365.57 - 87.52 | 242.62 - 80.05 | 313.35 - 79.92 | 357.47 - 119.82 |
| WUE (%) | 123.46 + 100.21 | 104.64 + 85.25 | 110.13 - 72.14 | 114.76 + 104.48 | 95.17 + 78.5 | 127.85 + 102.21 |
| Plant N uptake ($kgNm^{-2}y^{-1}$) | 7.72 - 0.47 | 2.17 + 0.09 | 1.86 - 0.31 | 7.24 + 0.02 | 1.14 + 0.12 | 1.32 - 0.29 |

**Table 3.** Mean values at the start of the 100-year experiment and net change by the end of the experiment for the prior model and the optimised models.

lead to higher starting values (except for WUE), and including data from FACE is found to induce a larger increase. When considering the total plant N uptake for both vegetation types, the prior starts with a high value and the optimised runs with much lower values.

In Fig. 6, we consider the net increase over the 100-year simulation. For both vegetation types, the rate of increase of GPP found by using parameters from the Fluxnet-only optimisation is the highest. For TeNE, the curve is starting to plateau due to a progressive nitrogen limitation. However, in both cases, the rate of increase is much faster compared to the runs with parameters from the $Flx_{GN}$-BOTH optimisation. The fertilisation rate decreases compared to the Fluxnet only optimisation. In Fig. 2, we saw that the $Flx_{GN}$-BOTH optimisation was starting to model the declining productivity response at ORNL. We may still be overestimating the fertilisation effect over TeBS sites and we could expect an even stronger response to nitrogen limitation - further reducing this GPP increase. Although the GPP increase is similar to the prior model for forest types, the larger spread among sites for $Flx_{GN}$-BOTH is notable - with negative GPP change at some sites suggesting a stronger nitrogen limitation under these parameters.

For transpiration rates over both TeBS and TeNE, the $Flx_{GN}$ simulations result is the smallest change by the end of the century. In contrast, the simulations $Flx_{GN}$-BOTH result in the largest change. For WUE, the differences observed in GPP and transpiration are mostly cancelled out and therefore, the different parameters do not elicit a great variation in responses. For TeBS, the prior and $Flx_{GN}$ simulations give a similar increase, with $Flx_{GN}$-BOTH suggests a slightly weaker increase. For TeNE, it is the opposite - $Flx_{GN}$ is slightly weaker than the other two simulations.

For plant N uptake, changes over the 100-year period are small in magnitude but vary between the different optimisations. The prior model shows a range of responses over the different sites (evidenced by the large spread), overall decreasing the rate of N uptake for TeBS and increasing the rate of uptake for TeNE. The Fluxnet-only optimisations lead to a slight increase in plant N update by the end of the century for both vegetation types. In contrast, the FACE optimisation lead to a decreased plant N update. This is especially notable for TeNE where the spread of responses over sites is greatly reduced. In all cases, changes to the rate of change occur during the first half of the simulations, plateauing to a constant value for the rest of the runs.

Note that our experiment only looks at increasing $CO_2$ while keeping the other drivers constant. It is possible that we would see different responses if we were to include changes in meteorological forcing (to mimic climate change) or changes

in nitrogen deposition and fertilisation, which would change nitrogen limitation and responses to water stress. Although we performed a sensitivity analysis to select the parameters, it is possible that additional parameters (e.g., ones controlling water stress or controlling the allocation of carbon in the plant) will give a different response. Furthermore, direct structural changes or addition to the model code could also change the results. Nevertheless, the fact that different parameters give such a varied model response to elevated $CO_2$ for a given model structure highlights the importance of finding a robust set of parameters to have faith in. When performing the Fluxnet optimisations, the optimisation against GPP and NEE gave the best fit to both quantities resulting in sets of parameters that worked well across each PFT. However, when considering production through NPP and LAI under different atmospheric $CO_2$ conditions, we found that the parameters were unable to capture the differences observed in data under different $CO_2$ regimes. Instead, the parameter sets found by optimising the model against Fluxnet data and both ambient and elevated $CO_2$ conditions were found to be more robust. These parameters resulted in the best fits overall against NPP and LAI under two different $CO_2$ regimes. As such, we have more confidence in these parameters and their ability to simulate terrestrial production under different atmospheric $CO_2$ conditions, leading us to have more faith in the model projections performed with them.

## 4  Conclusions

As our terrestrial models become more complex through the addition of more processes, we need to confront them with observed data to ensure we have confidence in the model's predictions. Manipulation experiments allow us to test the model under different $CO_2$ regimes and its capabilities to reproduce the ecosystem responses. FACE sites in particular are an important tool in evaluating modelled ecosystem response to climate change. They can be thought of as space-for-time substitution experiments (Rastetter, 1996), but where the change in atmospheric $CO_2$ is controlled and even manipulated to exceed conditions naturally found around the globe currently. By optimising model parameters against data from both ambient and elevated atmospheric $CO_2$ conditions, we gain confidence in the model's ability to reproduce fluxes under different atmospheric conditions. It will be interesting to use these parameters further to assess the evaluation of carbon stocks under high concentrations of $CO_2$ and at a larger scale to evaluate more directly the impact on the global sink.

We find that through the different optimisations of this carbon-nitrogen version of ORCHIDEE, we are able to improve the representation of simulated productivity. All optimisations are able to improve modelled GPP, and we generally improve the magnitude for NPP for the two levels of atmospheric $CO_2$. However, we do not achieve as good an improvement against respiration and, therefore, against NEE. Although we are unable to capture fully the inter-annual variability of NPP after optimisation, at ORNL we do start to model a negative trend for the NPP ratio, which is apparent in the observations but not simulated in the prior model. This suggests that the optimised parameters are able to capture the progressive nitrogen limitation at this site. We do struggle to capture the seasonal cycle of LAI properly at both FACE sites, suggesting incorrect LAI allocation. These results highlight the fact that optimising an LSM with the nitrogen cycle is more difficult and complex than with a carbon-only LSM, given the increased model feedbacks. In particular the dependence of plant productivity on soil nitrogen availability and in reverse the dependence of soil N content to litter input (hence productivity) induce strong

positive feedbacks. This provides a warning to other modelling groups looking to calibrate the carbon and nitrogen cycles in their model. Overall the current optimisation performs slightly worse than compared to the prior optimisations of previous ORCHIDEE versions (e.g., Kuppel et al., 2012). One example of increased model feedbacks is through the KSoil parameter. Without the nitrogen cycle, by changing the initial carbon stocks, this parameter was able to fix the magnitude of the respiration flux. However, the parameter now also changes the initial nitrogen stock and hence the mineralisation flux in the soil, which

impacts GPP. Another approach would have been to have several multiplicative factors each changing different pools or indeed one for each C and N. However, this would likely lead to more complications given the strong feedbacks observed. If one pool declines more than others in terms of C and N content, or if one pool becomes more depleted in N, it is probable that the model will enter a transient phase with potentially strong compensating fluxes, such as a large net carbon flux, to restore internal consistency. Ideally, we would optimize various model parameters that govern the turnover time and C:N ratio of each

pool throughout the entire spin-up period. However, achieving such optimization is not currently feasible.

Although the optimisation is not as optimal as that achieved with a carbon-only model, this work opens a new avenue to validate LSMs quantitatively with FACE data. We see that not only is there a benefit of adding FACE data on top of Fluxnet data when optimising a land surface model, it is, in fact, risky not to. The Fluxnet-only optimisations do not perform well under elevated conditions, which is critical when predicting the terrestrial response to climate change. Furthermore, since we see

that the future evolution of terrestrial productivity change is sensitive to the parameter values used in the model, getting these parameters right is critical. This is notable for both vegetation types where the Fluxnet-only optimisations and the optimisations with FACE data give very different trajectories in the idealized 1% $CO_2$ experiments, with the Fluxnet-only optimisations likely overestimating the $CO_2$ fertilisation effect. However, we do need to be cautious in assessing these results since we are only using one FACE site for each PFT meaning we are likely tuning to the specificities of that site. For example, ORNL shows a

progressive nitrogen limitation but this is not expected over all sites. Ideally, we would include a lot more FACE sites to capture different conditions. Especially, if we could optimise by grouping sites based on different levels of nitrogen limitation, then if the posterior parameters were found to be similar then the model processes allow for these differences.

In any optimisation, there is always a danger of overfitting to data limiting the generalisability of the calibrated model. By optimising the model against a number of different constraints (i.e. more than one data stream), we decrease the risk of

495 overfitting and therefore, gain some confidence in our parameter and hence in the projections. Such experiments can help us to describe better the future fertilizing effect of $CO_2$ under possible nitrogen limitation. However, we find that in our study, two sites are not sufficient to draw such conclusions about terrestrial responses to elevated $CO_2$, which could vary over different ecosystems. Although we have shown this approach of joint optimisations to be promising, more sites are needed. It would also be interesting to have different levels of nitrogen input at these sites to asses more clearly the nitrogen limitation on the $CO_2$

fertilization effect. We are also limited by the data the sites can provide. Due to the $CO_2$ fumigation over FACE sites, daily NEE cannot be measured at FACE sites, and therefore, GPP and TER estimates cannot be derived. These variables are more directly linked to productivity than leaf area index, the variable we use in our optimisations. In future optimisations, we might also want to include more 'nitrogen-type variables in the cost function, such as leaf nitrogen content. There are other processes at play that also need to be assessed. For example, the effect of soil moisture and the stress response to water availability will

also impact the mineralisation of organic matter and, thus, nutrient availability. Finally, structural changes do need to be made to the model to better capture the inter-annual variability of simulated NPP and LAI. This highlights how we can use FACE data to identify structural issues in models providing an important tool for model development.

Identifying structural deficiencies is the main strength of parameter estimation - we need to be sure that we are not simulating the right model output for the wrong reasons. Indeed, if we had not been able to find a set of parameters that gave a satisfactory fit to both atmospheric regimes, this would have highlighted a missing process in the model. Ideally, we would want to calibrate under ambient and test the robustness of the theory under elevated $CO_2$; however, given all missing (e.g., adaptation) or highly simplified processes (nutrient limitation), using both conditions is one approach to improve the overall model behaviour while highlighting these deficiencies. Although not shown, our framework also allows us to compute the posterior parameter uncertainty, which again can be very informative for model development. We do not discuss them in this paper since our imperfect setup (i.e., diagonal R matrix) means the information content of the observations is overestimated in the optimisation, but we do find that the uncertainty parameters are strongly reduced in all cases.

Although we performed a sensitivity analysis to select sensitivity model parameters, a large number of parameters were kept, some of which are indirectly impacted by the processes optimised. This can pose a risk, since changing such parameters may have an important impact elsewhere in the model and, therefore, may result in a degradation in model-data fit against processes not considered in the optimisation. Therefore, this study is only a first step toward a more comprehensive approach with more data streams.

Nevertheless, although it is by no means exhaustive, this proof-of-concept experiment highlights the importance of manipulation experiments and the additional information they can provide for model improvement. This is the first study of a global process-based model using data in this way. With more FACE sites, these types of data could be used more consistently used as part of the model optimisation procedures. Other data streams, such as normalized difference vegetation index, solar-induced fluorescence satellite data, and tree rings, could also be used to complement such optimisations giving the best constraints on the model parameters and hence on future climate predictions.

*Code availability.* The source code for the ORCHIDEE version used in this model is freely available online via the following address: https://doi.org/10.14768/c429d5c4-1164-45a7-8556-9bbef31baee3, and the optimisation tool is available through a dedicated web site for data assimilation with ORCHIDEE (https://orchidas.lsce.ipsl.fr).

## Appendix A: Optimised parameters

All processes and equations of ORCHIDEE can be found in the different documenting publications (e.g., Krinner et al., 2005), as well as on its website (https://forge.ipsl.jussieu.fr/orchidee/wiki/Documentation/UserGuide, last accessed:23/08/23). Here, we only highlight the impacted modules, summarise the equations in which the optimised parameters feature, and cite the relevant publications. Parameters optimised in the study, listed in Table 1, are coloured red in the following text.

## A1  Nitrogen-related processes

The nitrogen-related parameters and their equations are thoroughly described in **?**. In this version of ORCHIDEE-CN, we prescribe leaf nitrogen concentrations. This means that leaf C/N ratio is fixed within a prescribed range given by two of our parameters ($[CN_{leaf, min}; CN_{leaf, max}]; g_C[g_N]^{-1}$). To account for the nitrogen limitation on photosynthetic activity, VCmax (the maximum rate of Rubisco-activity-limited carboxylation) becomes a function of the leaf nitrogen content ($N_l$) as proposed by Kattge et al. (2009):

$$VC_{max} = NUE_{opt} * N_l \tag{A1}$$

where $NUE_{opt}$ ($\mu$mol $CO_2 s^{-1}[g_{Nleaf}]^{-1}$) is the nitrogen-use efficiency.

$N_l$ decreases exponentially from the top to the bottom of the canopy with decreasing light levels or increasing canopy depth. The value of $N_l$ at the top of the canopy, $N_l(0)$, is expressed a function of total canopy N content, $N_{tot}$, and the LAI of the total canopy, $L_{tot}$:

$$N_l(0) = \frac{k_N N_{tot}}{1 - \exp(-k_N L_{tot})} \tag{A2}$$

where $k_N$ is a specific extinction coefficient. Note that this is different to the extinction coefficient $k$ used to calculate the light profile within the canopy, although both are optimised. As we decrease through the canopy, the value of $N_l$ at a cumulative LAI (L) is defined following Dewar et al. (2012):

$$N_l(L) = N_l(0)\exp(-k_N L). \tag{A3}$$

It is assumed that $N_l$ varies through the canopy due to variations in specific leaf area (SLA; i.e., leaf area divided by leaf mass), instead of variations in leaf nitrogen concentration which is kept constant. The SLA at the bottom of the canopy (SLAinit) is fixed and also optimised.

The model calculates the nitrogen required (GN$_{init}$, $gm^{-2}d^{-1}$) to satisfy the new carbon ($G_C$ ($gm^{-2}d^{-1}$) to the different reservoirs under the assumption that CN$_{leaf}$ does not vary (Zaehle and Friend, 2010).

$$GN_{init} = (FCN_l/CN_{leaf} + FCN_{root}/CN_{root} + FCN_f/CN_{fruit} + FCN_{wood}/CN_{sap})G_C \tag{A4}$$

where FCN$_i$ represents the fractions (unitless) of carbon allocated to leaf (l), roots (root), fruit (f) and sapwood or stalks (wood), and CN$_i$ represents the C/N ratios for the different biomass pools at the previous time step. FCN$_{root}$ and FCN$_{wood}$ are optimised in this study. R$_{leaf}$ and R$_{root}$ are the fractions of N retranslocated when shedding leaves and roots (f$_{trans}$ parameter in Zaehle and Friend (2010)). CTE$_{bact}$ is a parameter relating denitrifier bacteria activity to Soil Organic Matter.

Root density follows an exponential profile, with more roots located in the top soil layers. The root density profile parameter z defines the depth above which $\sim 65\%$ of roots are stored and use the calculate plant moisture availability (Krinner et al., 2005, Eq. A18). Finally VMAX$_{UPTAKE}$ is used to calculate plant N uptake (Zaehle and Friend, 2010, supplementary material Eq. 8).

## A2 Allocation

Allocation in ORCHIDEE-CN follows the formalisms of the OCN model Zaehle and Friend (2010), further described in Naudts et al. (2015), and respects the pipe model theory (Shinozaki et al., 1964). $K_{LAtoSA}$ (whose range $K_{LAtoSA,min}, K_{LAtoSA,max}$ is calibrated) is used to derive a scaling factor between leaf and sapwood mass:

$$d_l = K_{LAtoSA} \times m_w \times d_s \tag{A5}$$

where $d_l$ is the one-sided leaf area of an individual plant, $d_s$ is the sapwood cross-section area of an individual plant and $m_w$ is the water stress. Sapwood mass ($M_s$) and root mass ($M_r$) are related as follows (following Magnani et al. (2000)):

$$M_s = k_{\text{sar}} \times d_h \times M_r \tag{A6}$$

where the parameter $k_{\text{sar}}$ is calculated:

$$k_{\text{sar}} = \sqrt{(K_{\text{root}}/K_{\text{sap}}) \times (k_{\tau s}/k_{\tau r}) \times k_{\rho s}} \tag{A7}$$

where $K_{\text{root}}$ is the hydraulic conductivity of roots, $K_{\text{sap}}$ is the hydraulic conductivity of sapwood, $k_{\tau s}$ and $k_{\tau r}$ are the longevity of sapwood and root respectively, and $k_{\rho s}$ is the sapwood density.

## A3 Phenology

For the phenology parameters, we mostly refer to MacBean et al. (2015). The hotosynthetic efficiency of leaves depends on their age $L_{\text{age}}$. Using four separate age classes, biomass newly allocated to leaves goes into the first age class and leaf biomass, Bl, is then transferred from one class to the next based on a PFT-specific critical leaf age value, $L_{\text{agecrit}}$. In temperate deciduous broadleaf forests, leaf senescence is triggered when the monthly air surface temperature goes below a threshold temperature:

$$T_{\text{threshold}} = T_{\text{senes}} + C_1 T_l + C_2 T_l^2 \tag{A8}$$

where $T_l$ is the long-term mean annual air surface temperature and $T_{\text{senes}}$, $C_1$ and $C_2$ are PFT-dependent parameters. Once senescence has begun, a fixed turnover rate is applied, with trees losing their fine roots at the same rate as their leaves

$$\Delta B = B.\Delta t / L_{\text{fall}} \tag{A9}$$

where $\Delta t$ is the daily time step, $B$ is the total biomass and $L_{\text{fall}}$ is optimised.

## A4 Photosynthesis

Stomatal conductance ($g_s$) is coupled to leaf photosynthesis by the following equation:

$$g_s = g_0 + \frac{A + R_d}{C_i - C_{i*}} f_{\text{VPD}} \tag{A10}$$

where $g_0$ is the residual stomatal conductance when irradiance approaches zero, $A$ ($\mu mol CO_2 m^{-2} s^{-1}$) is the net assimilation rate, $C_i$ ($mol CO_2 m^{-2}$) is the intercellular $CO_2$ partial pressure, $C_{i*}$ is the $C_i$-based $CO_2$ compensation point in the absence of respiration ($R_d$) and $f_{VPD}$ is the function for approximal effect of leaf-to-air vapour pressure difference (VPD, kPa):

$$f_{VPD} = \frac{1}{1/(A_1 - B_1 VPD) - 1} \tag{A11}$$

The empirical factors $A_1$ (unitless) and $B_1$ ($kPa^{-1}$) are optimised in this work.

## A5    Respiration

$Q_{10}$ (unitless) used to calculate the temperature control of heterotrophic respiration:

$$c_T = \min(1, Q_{10}^{(T-30)/10}) \tag{A12}$$

where $T$ is the surface/soil temperature for the above/below-ground pools.

The growth respiration is calculated as a fraction of the remaining total biomass:

$$R_g = FRAC_{growthresp} \times \max(B - \Delta t \times \sum R_{m,i}, 0.2 \times B) \tag{A13}$$

where B is the total biomass, $\Delta t$ the time step (one day), and $FRAC_{growthresp}$ a fraction to be optimized.

*Author contributions.* PP and NR conceived of the experiment. LE performed the preliminary experiments. NV developed the nitrogen version of the ORCHIDEE. VB developed the data assimilation system, which NR adapted to FACE experiments. BG provided the FACE data and configuration files needed for ORCHIDEE. AL provided expertise first running ORCHIDEE-CN. NR ran the main experiments and generated the figures. All authors contributed to writing and editing the manuscript.

*Competing interests.* The author declare no competing interesting

*Acknowledgements.* This work used eddy covariance data acquired and shared by the FLUXNET community, including these networks: AmeriFlux, AfriFlux, AsiaFlux, CarboAfrica, CarboEuropeIP, CarboItaly, CarboMont, ChinaFlux, Fluxnet-Canada, GreenGrass, ICOS, KoFlux, LBA, NECC, OzFlux-TERN, TCOS-Siberia, and USCCC. The FLUXNET eddy covariance data processing and harmonization was carried out by the ICOS Ecosystem Thematic Center, AmeriFlux Management Project and Fluxdata project of FLUXNET, with the support of CDIAC, and the OzFlux, ChinaFlux and AsiaFlux offices. This work also data from the FACE experiments: Oak Ridge National Laboratory and DUKE Forest.

We would also like to thank the wider ORCHIDEE development team for developing and maintaining the ORCHIDEE land surface model. We acknowledge the support by the ESM2025 project. This project has received funding from the European Union's Horizon 2020 research and innovation programme under grant agreement No 101003536. We also acknowledge funding from the European Union's Horizon 2020 research and innovation programme under the Marie Skłodowska-Curie grant agreement No 101020076.

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
