# Peer review of "Using free air CO2 enrichment data to constrain land surface model projections of the terrestrial carbon cycle"

_EGUsphere, 2023_

## Author Comment (AC1)

**Reviewer 1**

Overall, I applaud the authors on the use of multi-observation streams, including manipulation experiments to explore the extent they can better optimise their LSM. The paper is logically put together and clearly written. I have a few potential issues but hopefully, nothing that would preclude this manuscript from being published with some revision. This is an exciting approach and you could see how it might be extended for a broader range of flux/satellite and manipulation experiments (e.g. https://onlinelibrary.wiley.com/doi/full/10.1111/gcb.16585).

We would like to thank the reviewer for his positive review and enthusiasm for our work. We do indeed hope that this work will open up a lot of exciting new avenues given the wide range of manipulation experiments available. We also thank the reviewer for the citation which has been added to the text (L42).

One issue I have with the aims in the intro relates to the extent it makes sense to optimise against both ambient and elevated conditions (your line 77)? The underlying driving principle of the FACE model intercomparison was that the models should be able to predict eCO2 response given existing parameterisations and underlying theory. If the parameters need to be re-optimised for eCO2 conditions, this is fine, but it implies the models lack the theory to predict these changes. For example, in De Kauwe et al. 2013, we tested whether the stomatal slope changes with eCO2 (Fig 6) and found it did not (i.e. if the theory is right the ambient parameterisation should allow you to predict the eCO2 response), supporting the point I just made. This is not to say it isn't a valid question to ask / test, but I think the text requires more nuance that we currently see. For example, to achieve the improved fit when eCO2 was considered, what params had to change? More on this point below.

The reviewer is correct that, in theory, we should be able to calibrate against ambient conditions and keep the elevated data for the evaluation since a robust model should be able to predict changes under different conditions. However, we know that our models are not perfect - there are a lot of missing processes, for example, ones linked to plant adaptation, as well as issues linked to scale representation, for example, leaf-scale in situ measurements versus kilometer-wide model resolution. Furthermore, our treatment of the nutrient limitation, although state-of-the-art, remains a very rough parameterisation of the nitrogen cycle. Therefore, we can not expect to be able to calibrate under ambient and have the model automatically work under elevated $CO_2$ conditions.

We strongly agree that we shouldn't have a separate elevated $CO_2$ parameter set. Note that at the end of the BOTH calibration, we only have one set of parameters - a parameter set that works best under both atmospheric regimes. In addition, note also the fact there is

such large uncertainty in the parameters, it is also possible to have the correct theory (i.e., equations) but the wrong parameters. When optimising the cost function, we find that our parameter space is full of minima (Kuppel et al., 2012; Bastrikov et al, 2018). This means that there can be different parameter sets giving similar fits but with wildly different parameter values. Therefore, it is recommended to perform multi-data stream calibrations to a) help smooth parameter space and b) make sure we don't overfit to one data source. By treating ambient and elevated conditions as two different data streams, we ensure we don't overfit to one regime but get the parameter set giving the compromise. Finally, given we will be using this model in CMIP-type exercises, we need to have the most operational version with a best set of parameters to use. As seen in Figure 5, for one model structure, different parameter values can give large variations in model responses. The proposed methodology in the paper (calibrating using data from both ambient and elevated $CO_2$ simultaneously) finds a parameter set we have more faith in than just calibrating against ambient conditions.

Nevertheless, the reviewer is correct that we do need to be cautious with our approach. The tuning is probably compensating for missing processes and wrong/un-precise parametrisations, as well incorrect model forcing. Parameter estimation can improve the model fit for the wrong reasons, hiding structural problems. However, it can also help us identify structural errors. Most of the time, we use this methodology to do just that - when we are unable to `match the observations using data assimilation, this helps us to identify structural deficiencies and therefore we feed back to the ORCHIDEE model developers to improve these deficiencies. For example, in this work, we found that, with the set of parameters optimised, we were unable to match the seasonality of LAI suggesting we need to reconsider how LAI phenology is modelled in ORCHIDEE. Furthermore, if we had been unable to find set of parameters that worked satisfactorily under both conditions, then this would have suggested a process was missing - like a functional response to $CO_2$. We have expanded the text on L443 to add this discussion to the conclusions:

"...Finally, structural changes do need to be made to the model to better capture the inter-annual variability of simulated NPP and LAI.

**Identifying structural deficiencies is the main strength of parameter estimation - we need to be sure that we are not simulating the right model output for the wrong reasons. Indeed, if we had not been able to find a set of parameters that gave a satisfactory fit to both atmospheric regimes, this would have highlighted a missing process in the model. Ideally, we would want to calibrate under ambient and test the robustness of the theory under eCO2; however, given all missing (e.g., adaptation) or highly simplified processes (nutrient limitation), using both conditions is one approach to improve the overall model behaviour while highlighting these deficiencies."**

We have also changed L77 to be clearer in the introduction:

> "Furthermore, by optimising a land surface model to both ambient and elevated conditions **simultaneously**, we will gain extra confidence in the model projections using this **single** set of parameters. **Although ideally we would want to calibrate under ambient conditions and test the model under elevated conditions, known model structural errors do not guarantee that the model is able to predict changes under different conditions. As such, we provide an alternative approach to model calibration, maximising the available information content of the optimisations.**"

Finally, we acknowledge that information about the posterior parameters is missing from the manuscript. Since both reviewers have highlighted this gap in our manuscript, we have added a section to discuss how the parameters change. More on this below.

A second issue I have relates to the breadth of parameters to be optimised and the data constraint. For example, to what extent can we expect to constrain a range of these quantities (e.g. litterfall, leafage, LAtoSA, etc) from gross or net carbon fluxes? Clearly indirectly a quantity like the leaf-2-sapwood area affects LAI, which affects GPP, but it feels quite downstream and I wonder why we think the gross flux would offer a strong (any) constraint here? Equally, if it turns out that it does (I'm writing this comment in advance of reading the results), what does this really mean? Again not a problem, but I think as a reader I'd appreciate a sentence or two in the methods on rationale or expectations here.

The reviewer is right to highlight this - this is often a problem in land surface model data assimilation. There are a lot of parameters we can calibrate - most of which will have an indirect effect on the flux assimilated. This is why we performed a sensitivity analysis study to first select the parameters. We acknowledge that this crucial step may be a bit lost in the manuscript since we have put this in the model/parameter section of the Methods and Data. As such we have added the following to better highlight the sensitivity study (and its motivation):

> In Section 2.4 "Performed experiments" L292:
>
>> "**Before performing the optimisations, we also conducted a sensitivity analysis on the parameters (as described in Sect. 2.1.2 and shown in Table 1). A sensitivity analysis tests how different the model outputs change with respect to different parameters. This was done to ensure that only parameters showing some sensitivity to the model outputs were used in the optimisation and therefore minimising the risk**

**of using parameters that are weakly constrained by the fluxes. This is an important step since we want to avoid constraining parameters that will have a small impact on the optimisation but have the potential to significantly degrade the model-data fit against processes not included in the calibration.**

Once spun up **and with the list sensitive parameters, …"**

In Section 3.1 "Fluxnet optimisations" L239:

"... and whether they can be optimised using observed data **and the sensitive parameters identified in the Morris experiment (Table 1)**."

Nevertheless, even after the sensitivity analysis, some parameters are kept that are indirect and therefore weakly constrained by NPP and LAI data. We recognised that this is a risk and that we may overtune parameters without other data. Note that in addition to the gross and net carbon fluxes used, we do assimilate against LAI data, which should provide a more direct constraint on the phonology parameters highlighted by the reviewer (i.e., leaf age crit and leaf-2-sapwood). Nevertheless, more data streams constraining specific processes would be better. This study is only a first step toward a more comprehensive complete calibration. In future, we may consider some sort of cascade tuning to use specific data directly related to specific processes at each step, however, this is beyond the scope of this work. We have added the following text to the conclusions to acknowledge this risk (L445):

**"Although we performed a sensitivity analysis to select sensitivity model parameters, a large number of parameters were kept, some of which are indirectly impacted by the processes optimised. This can pose a risk, since changing such parameters may have an important impact elsewhere in the model and, therefore, may result in a degradation in model-data fit against processes not considered in the optimisation. Therefore, this study is only a first step toward a more comprehensive approach with more data streams."**

A third issue and one to tackle in the discussion relates to how best to utilise FACE data. Overall, I applaud the approach taken here by the ORCHIDEE group but equally have concerns that an implication of calibrating to ORNL for *all* broadleaved sites could inadvertently impose a progressive nitrogen limitation (as observed at this experimental site) across sites where this may not be true. I think it is important to predict the response at ORNL for the right reasons (see Zaehle et al. 2013 New Phyt) rather than using this as a basis for all nitrogen-limited sites. I'd go as far as to say it may even make sense *not* to well match the decline in NPP at this site from a standard tuning exercise.

It is true that we do need to be careful since we do not want to impart the wrong information on the Fluxnet sites. The best way to do this would be to include a lot more FACE sites to capture different conditions. Especially, if we could optimise by grouping sites based on different levels of nitrogen limitation, then if the posterior parameters were found to be similar then the model processes allow for these differences. This study demonstrates the untapped potential of FACE data - but is not without its limitations. We have expanded on the need for more FACE sites in the conclusion (L447):

> "...Fluxnet-only optimisations likely overestimating the $CO_2$ fertilisation effect. **However, we do need to be cautious in assessing these results since we are only using one FACE site for each PFT meaning we are likely tuning to the specificities of that site. For example, ORNL shows a progressive nitrogen limitation but this is not expected over all sites. Ideally, we would include a lot more FACE sites to capture different conditions. Especially, if we could optimise by grouping sites based on different levels of nitrogen limitation, then if the posterior parameters were found to be similar then the model processes allow for these differences.**"

The fourth issue that I felt was lacking from the results was the link between what was tuned and the emergent results. To improve the fluxes are a similar set of params being adjusted for PFT group? Or to obtain better fluxes are the tuned params bespoke per site? Apart from the discussion about Ksoil I'm entirely unclear how and by how much the parameters have changed, this relates to my "second issue" above. As a reader, we do not get any sense of which of the params in table 1 are most important, effectively the source of the improvement is opaque. I'm left to ponder if the tuned params made logical sense or if they reflect a repartitioning of model error or uncertainty.

As mentioned in response to the "second issue", we have added the following section which adds more transparency about the parameter changes.

**3.2.3 Posterior parameter values**

In Fig. 3, we consider how the different parameters have changed by comparing results from the Fluxnet-only optimisation and the optimisation including the FACE data. For the temperate broadleaf summergreen parameters it is hard to distinguish significant patterns. The posterior values sometimes move in the same direction for both optimisations (e.g., $K_N$, $FCN_{wood}$, $R_{root}$ and z), other times the parameters are pulled in different directions (e.g., $CN_{leaf,max}$ and $R_{leaf}$). When considering the most sensitive parameters, both optimisations agree with the direction of change, with the exception of $NUE_{opt}$. For the photosynthetic parameters $A_1$ and $B_1$, we see that

these are changed during the Fluxnet-only optimisation but are less important in correcting the fit when the FACE data is included in the optimisation. Overall this highlights that adding the FACE data can pull the parameters in a different direction than when using only Fluxnet data. This indicates that there are likely compensating effects (or different repartitioning of model error). It is possible that with more data sets, the results would be more robust but this would need to be confirmed with the use and assimilation of additional data streams.

In contrast, for the temperate needleleaf parameters, nearly half (11/23) of the parameters were unchanged (or only slightly) during the Fluxnet-only optimisation. However, when optimised with the additional FACE data, these parameters changed greatly. These parameters include a number of the most sensitive parameters suggesting these are especially important in capturing the model response under both atmospheric regimes for needleleaf sites.

Although looking at these parameter values can be very informative - we must remember that there are complex interactions between the parameters and processes that will not be evident by just looking at these values.

[Figure]

**Figure 3**: Posterior parameter values of the Fluxnet-only optimisation (blue, $Flx_{GN}$) and the optimisation incorporating FACE data under both atmospheric regimes (orange, $Flx_{GN}$-BOTH). The horizontal line represents the prior value of each parameter and each box spans the range of variation

allowed for each parameter during the optimisation. Parameters highlighted in bold correspond to the parameters identified as most sensitive in the preliminary Morris experiment (see Table 1).

Fifth, in the discussion of Figure 5, you might be able to link to the Walker 2015 paper (https://agupubs.onlinelibrary.wiley.com/doi/10.1002/2014GB004995), there they also imagined what would happen to models if ORNL and duke continued for 300 years. This is similar to your idealised experiment. In particular, that paper extracts the change in N availability over time, which could be something you potentially examine in your analysis. Currently, when you optimise against FACE vs Fluxnet, you get a much lower GPP response, which leads me to wonder how you've affected N cycling cf. Walker et al.

We thank the reviewer for the suggestion. Although it is hard to relate the different "GPP vs. CO2" responses obtained for the different sets of optimizations to potential impacts on the N cycle, and so directly compare to Walker et al, we are able to include the evolution of plant N uptake in Figure 5. The difficulty in performing such a comparison arises because similar changes in responses might be attributed to different parameter adjustments, and different parameters do not impact the N cycle in the same way. Nevertheless, by examining the time evolution of plant N uptake in Figure 5 we can provide valuable insights into how the N uptake varies according to different optimization approaches. Figure 5 and Table 3 have been expanded to include Plant N uptake change as follows, and the text expanded to discuss these results:

| | TeBS | | | TeNE | | |
|---|---|---|---|---|---|---|
| | Prior | Optimised | | Prior | Optimised | |
| | | Fluxnet only | Fluxnet & FACE | | Fluxnet only | Fluxnet & FACE |
| GPP (kgCm$^{-2}$y$^{-1}$) | 5.22 + 0.55 | 3.58 + 1.95 | 4.03 + 0.99 | 2.85 + 0.73 | 3.93 + 1.73 | 4.59 + 0.66 |
| Transpiration (mm.y$^{-1}$) | 417.29 - 83.43 | 342.93 - 47.06 | 365.57 - 87.52 | 242.62 - 80.05 | 313.35 - 79.92 | 357.47 - 119.82 |
| WUE (%) | 123.46 + 100.21 | 104.64 + 85.25 | 110.13 - 72.14 | 114.76 + 104.48 | 95.17 + 78.5 | 127.85 + 102.21 |
| Plant N uptake (kgNm$^{-2}$y$^{-1}$) | 7.72 - 0.47 | 2.17 + 0.09 | 1.86 - 0.31 | 7.24 + 0.02 | 1.14 + 0.12 | 1.32 - 0.29 |

**Table 3.** Mean values at the start of the 100-year experiment and net change by the end of the experiment for the prior model and the optimised models.

"…The optimisations lead to higher starting values (except for WUE), and including data from FACE is found to induce a larger increase. **When considering the total plant N uptake for both vegetation types, the prior starts with a high value and the optimised runs with much lower values.**"

[Figure]

"For plant N uptake, changes over the 100-year period are small in magnitude but vary between the different optimisations. The prior model shows a range of responses over the different sites (evidenced by the large spread), overall decreasing the rate of N uptake for TeBS and increasing the rate of uptake for TeNE. The Fluxnet-only optimisations lead to a slight increase in plant N update by the end of the century for both vegetation types. In contrast, the FACE optimisation lead to a decreased plant N update. This is especially notable for TENE where the spread of responses over sites is greatly reduced. In all cases, changes to the rate of change occur during the first half of the simulations, plateauing to a constant value for the rest of the runs."

We have also added the following to the conclusions to highlight the perspective of constraining more directly the N cycle in future optimisations (L441):

These variables are more directly linked to productivity than leaf area index, the variable we use in our optimisations. **In future optimisations, we might also want to include more 'nitrogen-type variables in the cost function, such as leaf nitrogen content.**

Finally, the question that I wondered as I was reading the results: "to what extent the calibration against flux + FACE improves the estimated fluxes in Figure 1 (or not)?". I think section 3.2 & Fig 4 addresses this question but I got a bit lost as to whether the multi-site calibration was then being used to re-examine the flux predictions or not.

This is indeed what we show in Section 3.2.3 and Fig. 4. We agree that this section would benefit from stronger callbacks to Figure 1 and text clarify this. As such, we have added the following:

    To L343:

        **"In other words, this helps us see to what extent the calibration with FACE data changes the estimated fluxes shown in Fig. 1."**

    Caption of figure 4

        **"This figure complements Fig. 1, here showing the optimisations using the FACE data fit the timeseries compared to the prior model and the optimisation (GN) without FACE."**

Minor

=====

- Line 54. "experiment" - suggest you change it to experiment"s" or "one such type of experiment", currently the text implies there is only a single FACE experiment

Agreed, this has been fixed in the text.

- Line 57-59. I didn't really get the point of these two sentences, they aren't linked at all to the text. There have been various other reviews of FACE which you don't list but could do. I think you either need to link this to the existing text or omit these lines.

We have chosen to simply omit these lines to avoid confusion.

- Line 203. While I do take the argument being made here, I do wonder if it makes sense to calibrate against all of the fluxes. I feel like you would calibrate against TER and NEE and perhaps the assessment would be on how GPP would change. I don't necessarily have a recommended change or response needed here, but it feels intuitively wrong to calibrate against all three fluxes. The authors are far more the experts in this field though so it is up to them of course.

Apologies for this not being clearer - we never optimise against all three. We either do TER and NEE or GPP and NEE. However, the three fluxes are used in the evaluation. This has been clarified in the text:

    **"In each case, two fluxes are used in optimisations."**

Martin De Kauwe

---

## Author Comment (AC2)

**Reviewer 2**

Raoult describes a model parameter tuning exercise using the ORCHIDEE model and Fluxnet + CO2 FACE experiments. It builds on recent work comparing model simulations to global change experiments by directly tuning the model to FACE experiments for two PFT types.

We would like to thank the reviewer for taking the time to read and comment on our manuscript.

The take-home messages are that the parameter tuning with the FACE studies seems to improve some aspects of the model and not others and that tuning to the flux data alone can alter the predicted response to rising CO2 (I did find it interesting that the prior and the Flux+FACE predictions in Figure 5 were similar to the priors, suggesting knowledge of how the model responded to CO2 likely influence past parameter sets).

Yes, this is interesting but we cannot be sure whether this is due to prior knowledge or just a coincidence - an artefact of this model version - since posterior parameter values are different. It speaks more to the fact that there are a lot of possible parameter sets, which may give similar response in some parts of the model and very different response in other parts.

I appreciate the inclusion of the FlxGN-AMB simulation because without its inclusion it would be hard to separate the influence of the type of data at the FACE sites (NPP, LAI) from the use of the experiments.

I would challenge the authors to make the manuscript have more impact beyond users of the ORCHIDEE model. What can other modeling groups learn from the manuscript? As an example, the manuscript discusses how the inclusion of the nitrogen cycle influenced the capacity to tune the model to the FACE studies. However, this is not directly tested in the results in a standardized framework. Adding those results would alert other modeling groups to be cautious with optimizing the C + N model using a similar approach as a C-only model. Related, are there any suggestions about how to update the parameter tuning to work better with N cycle models. It seems the KSoil parameter was an issue because it adjusted both C and N pools. Should there be a KSoil for each soil pool so that pools that have high N mineralization rates are adjusted differently than ones with lower N mineralization rates?

We agree that we do need to better highlight the broader perspectives of this paper and how the wider modelling community can benefit from this study. As discussed in the

conclusions, we present not only methodology for optimisation against these data but also a warning to other modelling groups looking to calibrate their model, especially since most land surface models are now starting to include a nitrogen cycle. We have added the following to L420:

> These results highlight the fact that optimising an LSM with the nitrogen cycle is more difficult and complex than with a carbon-only LSM, given the increased model feedbacks. **In particular the dependence of plant productivity on soil nitrogen availability and in reverse the dependence of soil N content to litter input (hence productivity) induce strong positive feedbacks. This provides a warning to other modelling groups looking to calibrate the carbon and nitrogen cycles in their model.**

We also highlight a new use for FACE data, which importantly can be used to identify structurally deficiencies in the model (L444):

> Finally, structural changes do need to be made to the model to better capture the inter-annual variability of simulated NPP and LAI. **This highlights how we can use FACE data to identify structural issues in models providing an important tool for model development.**

One of the biggest lessons from the study was the added complexity of getting the initial carbon pools right now that we need to maintain C/N ratio. We choose to implement one multiplicative scalar (KSoil) to the different pools, applying it to the slow, labile and active pools for both C and N content, thus preserving the C:N ratio of the pools. Using a different KSoil for each pool or a different KSoil for C and N, although technically feasible, would likely be associated with adverse effects. If one pool decreases more than the others (both for C and N content) or one pool becomes more depleted in N than the others at the initial time step, it is likely that the model will enter a transient phase with possibly strong compensating fluxes (i.e., possibly a large net carbon flux) in order to reach again the internal model consistency. Ideally, we would need to optimize the different model parameters controlling the turnover time of each pool and their C:N ratio over the full duration of the spin-up, which is practically not feasible. We have added the following to the conclusions (L424):

> However, the parameter now also changes the initial nitrogen stock and hence the mineralisation flux in the soil, which impacts GPP. **Another approach would have been to have several multiplicative factors each changing different pools or indeed one for each C and N. However, this would likely lead to more complications given the strong feedbacks observed. If one pool declines more than others in terms of C and N content, or if one pool becomes more depleted in N, it is probable that the model will enter a transient phase with potentially**

**strong compensating fluxes, such as a large net carbon flux, to restore internal consistency. Ideally, we would optimize various model parameters that govern the turnover time and C:N ratio of each pool throughout the entire spin-up period. However, achieving such optimization is not currently feasible.**

Specific comments:

1. More description of the ORNL and Duke FACE data is needed.  How did you handle the split-plot design at Duke FACE?  Where specifically did the data come from? Pulled from the table of a manuscript or from a data repository?

   We have added the following to Section 2.3 "In situ data"

   > "**The data for these sites came from the FACE-MDS project (Walker et al., 2018a and 2018b; https://facedata.ornl.gov/facemds/).** For each site, we used the data from two experimental plots **(with their associated error bars)**; one with unaffected atmospheric $CO_2$, i.e. ambient (AMB), and one with elevated atmospheric $CO_2$ (ELE). **Although the DUKE experiment also has ammonium nitrate treatments at half of its plots from 2005 onwards (Feng et al., 2010), we only consider the data from the plots without nitrogen fertilization.**"

   References:

   FACE-MDS Phase 2: Meteorological Data and Protocols. Walker, A.P., Yang, B., Boden, T., De Kauwe, M.G., Fenstermaker, L.F., Medlyn, B., Megonigal, J.P., Oren, R., Pendall, E., Zak, D.R., Zaehle, S., Burton, A.J., Drake, B.G., Evans, R.D., Hungate, B., Johnson, D.P., Kim, D., LeCain, D., Lewin, K.F., Lu, M., Mueller, K.F., Nowak, R.S., Riggs, J.S., Smith, S.D., Tharp, L.M., Zelikova, T.J., Norby, R.J., 2018a. doi:10.15485/1480325

   FACE-MDS Phase 2: Data from Six US-Located Elevated $CO_2$ Experiments. Walker, A.P., De Kauwe, M.G., Fenstermaker, L.F., Hungate, B., Medlyn, B., Megonigal, J.P., Oren, R., Pendall, E., Talhelm, A.F., Zaehle, S., Zak, D.R., Boden, T., Brown, A.L., Burton, A.J., Butnor, J.R., Day, F.P., Drake, B.G., Dijkstra, P., Evans, R.D., Finzi, A.C., Iversen, C.M., Jackson, R.B., LeCain, D., McCarthy, H.R., Powell, T.L., Nowak, R.S., Riggs, J.S., Smith, S.D., Stover, D.B., Tharp, L.M., Warren, J.M., Wullschleger, S.D., Norby, R.J., 2018b. doi:10.15485/1480325

Feng, X., Simpson, A. J., Schlesinger, W. H., & Simpson, M. J. (2010). Altered microbial community structure and organic matter composition under elevated CO2 and N fertilization in the duke forest. *Global Change Biology*, *16*(7), 2104-2116.

2. The use of parameter priors isn't clear. In standard Bayesian statistics, priors have a prior distribution. I think this prior distribution is a combination of the mean in Table 1 and the B matrix. A description of how the B matrix is created seems to be missing.

We define the prior distribution of each parameter to be a Gaussian spanning 40% of the prior range - which itself it elicited from expert knowledge. This description has been added to the text as follows (L149:

"**We define the prior distribution of each parameter to be 40% of the prior range.**"

3. The posterior parameter values should be added to Table 1 for the parameters that were fit.

We agree that the posterior parameter values are lacking in the manuscript. Instead of adding the values to Table 1 (which would complexify the table), we have added a figure illustrating how the parameters change, as well as a section discussing some of these changes.

4. How is the R matrix determined?

As in most studies, we set the **R** matrix to be diagonal. We defined the observation error (variance) as the mean-squared difference between the observations and the prior model simulation so that this variance reflects not only the measurement errors but also the model errors. Added the following to the text (L149):

" **We set both matrices to be diagonal. For B … For R, we defined the observation error (variance) as the mean-squared difference between the observations and the prior model simulation so that this variance reflects not only the measurement errors but also the model errors.**"

5. I recommend adding a discussion of the results in the context of Rastetter, E. B. (1996). Validating Models of Ecosystem Response to Global Change. *BioScience*, *46*(3), 190–198. https://doi.org/10.2307/1312740.

We thank the reviewer for sharing this paper - it will definitely help strengthen the discussion. The discussion has been expanded as follows (L406):

Manipulation experiments allow us to test the model under different $CO_2$ regimes and its capabilities to reproduce the ecosystem responses. **FACE sites in particular are an important tool in evaluating modelled ecosystem response to climate change. They can be thought of as space-for-time substitution experiments (Rastetter, 1996), but where the change in atmospheric $CO_2$ is controlled and even manipulated to exceed conditions naturally found around the globe currently.** By optimising model parameters against data from both ambient and elevated atmospheric

6. Line 436: the sentence talks about how the study reduces parameter uncertainty but the manuscript doesn't actually present the prior vs. posterior parameter uncertainty or any ensemble of simulations with different parameter values from the prior and posterior distributions. The study optimizes the parameter value to be more consistent with the observations but doesn't necessarily reduce the uncertainty.

It is true that we do not present the posterior uncertainties in this manuscript. We have calculated them using the Hessian at the optimal and found that the uncertainty is significantly reduced for all parameters. However, we are wary to add this information to the manuscript since we know our optimisation setup is not perfect, and while the posterior parameter values will be relatively unaffected by this, this can lead to complications when interpreting the posterior uncertainty reduction. Most notably, we do not include error correlations in the **R** matrix (note **R** is used in the calculation of the Hessian). This is because these correlations are extremely hard to quantify. Instead, we usually inflate the variances to account for the fact we are overestimating the information content of the observations. For this proof of concept study, we did not do this step - nevertheless, we did use large variances (defined by the mean squared difference between the model and observations) to partly account for this. In future studies, we would want to tune this multiplicative factor to ensure that the information content of the observations was not overestimated (using a $\chi^2$ test) or, in an ideal world, be able to quantify the off-diagonals of the **R** matrix.

We have changed the sentence to be more nuanced removing the mention of uncertainty:

"By optimising the model against a number of different constraints, we **gain confidence in our** parameter and hence in the projections"

And added the following to the text (L444):

"Finally, structural changes do need to be made to the model to better capture the inter-annual variability of simulated NPP and LAI. This highlights how we can use FACE data to identify structural issues in models providing an important tool for model development. **Although not shown, our framework also allows us to compute the posterior parameter uncertainty, which again can be very informative for model development. We do not discuss them in this paper since our imperfect setup (i.e., diagonal R matrix) means the information content of the observations is overestimated in the optimisation, but we do find that the uncertainty parameters are strongly reduced in all cases.**"

---

## Author Response (AR1)

Dear Dr. Raoult,

The two reviewers are generally supportive of the research presented in your manuscript. Your responses to the reviewers' comments and suggested edits are mostly convincing. However, I have some reservation about the following points raised by the reviewers (and some of my own) and your replies:

Thank you for taking the time to read the manuscript, the reviews and our responses. We have addressed your following concerns below.

**Some fundamentals of the methods are unclear or could be mis-understood:**

- Eq. 1 is in a matrix-form, but covariances are not considered. Can Eq. 1 be provided in a simplified form that also clarifies the weighing of error terms from evaluating against FACE and FLUXNET data separately?

The equation as given in the text is the notation commonly used in our the field and although we do not consider covariances here, in future, this is something we would like to move towards. Nevertheless, we agree that in this case it is also helpful including a second equation showing the decomposition of terms obtained through a diagonal R matrix:

> … For **R**, we defined the observation error (variance) as the mean-squared difference between the observations and the prior model simulation so that this variance reflects not only the measurement errors but also the model errors. **Furthermore, since we do not consider error covariances, R is diagonal and therefore we can decompose the first term of Eq. 1 into different terms for each assimilated datastream:**
>
> $J(\mathbf{x}) = k_{Flx}*(M_{Flx}(\mathbf{x}) - \mathbf{y}_{Flx})^{T}(1/\sigma_{Flx})(M_{Flx}(\mathbf{x})-\mathbf{y}_{Flx}) + k_{FACE}*(M_{FACE}(\mathbf{x}) - \mathbf{y}_{FACE})^{T}(1/\sigma_{FACE})(M_{FACE}(\mathbf{x})-\mathbf{y}_{FACE}) + (\mathbf{x} - \mathbf{x}_b)^{T}\mathbf{B}^{-1}(\mathbf{x}-\mathbf{x}_b)$
>
> **where Flx and FACE subscripts are used to denote the FLUXNET and FACE parts of the equation; $k_i$ denotes the weighting using for each datastream, $\sigma_i$ denotes the observational error, and $M_i$ and $y_i$ denote modelled and observed data streams.**

We add also added links to $k_{Flx}$ and $k_{FACE}$ in Sect. 2.4. (e.g. L232).

- Your use of the term 'data assimilation' is generous. I feel like 'parameter optimization' is a more appropriate term to be used here, since you are not rigorously estimating prediction uncertainties (you write that this manuscript is a "proof of concept study" in your reply to reviewer 2), and it is unclear how observation errors are treated.

The use of "Data Assimilation" for parameter optimisation is common our field of study, and the inclusion of the KSoil parameter allows us for perform a station optimisation of the initial soil C poosl. Nevertheless, we have softened the language and used parameter estimation instead of the data assimilation for a better clarity and accessibility.

- The calibrated parameters are insufficiently described, no equations where and how they are used are provided, and their units are not specified in Table 1.

We have added the units to Table 1 and equations in Appendix. We have added the following to the text to relax this:

[revised manuscript text omitted]

$$d_1 = K_{latosa} \times m_w \times d_s$$

where $d_l$ is the one-sided leaf area of an individual plant, $d_s$ is the sapwood cross-section area of an individual plant and $m_w$ is the water stress. Sapwood mass ($M_s$) and root mass ($M_r$) are related as follows (following Magnani et al., 2000):

$$M_s = k_{sar} \times d_h \times M_r$$

where the parameter $k_{sar}$ is calculated:

$$k_{sar} \sqrt{(k_{root}/k_{sap})} \times (k_{TS}/k_{Tr}) \times k_{\rho s}$$

where $k_{root}$ is the hydraulic conductivity of roots, $k_{sap}$ is the hydraulic conductivity of sapwood, $k_{TS}$ is the longevity of sapwood and $k_{Tr}$ is the root longevity, and $k_{\rho s}$ is the sapwood density.

**Phenology**

For the phenology parameters we mostly refer to MacBean et al., (2015). The photosynthetic efficiency of leaves depends on their age $L_{age}$. Using four separate age classes, biomass newly allocated to leaves goes into the first age class and leaf biomass, $B_l$, is then transferred from one class to the next based on a PFT-specific critical leaf age value, $L_{agecrit}$. In temperate deciduous broadleaf forests, leaf senescence is triggered when the monthly air surface temperature goes below a threshold temperature:

$$T_{threshold} = T_{senes} + C_1 T_l + C_2 T_l^2$$

where $T_l$ is the long-term mean annual air surface temperature and $T_{senes}$, $C_1$ and $C_2$ are PFT-dependent parameters. Once senescence has begun, a fixed turnover rate is applied, with trees losing their fine roots at the same rate as their leaves

$$\Delta B = B.\Delta t/L_{fall}$$

where $\Delta t$ is the daily time step, B is the total biomass and $L_{fall}$ is optimised.

**Photosynthesis**

Stomatal conductance ($g_s$) is coupled to leaf photosynthesis by the following equation:

$$g_s = g_0 + \frac{A + R_d}{C_i - C_{i*}} f_{VPD}$$

where $g_0$ is the residual stomatal conductance when irradiance approaches zero, A ($\mu$ mol $CO_2$ m$^{-2}$s$^{-1}$) is the net assimilation rate, $C_i$ (mol $CO_2$m$^{-2}$) is the intercellular $CO_2$ partial pressure, $C_{i*}$ is the $C_i$-based $CO_2$ compensation point in the absence of respiration ($R_d$) and $f_{VPD}$ is the function for the approximal effect of leaf-to-air vapour pressure difference (VPD, kPa):

$$f_{VPD} = \frac{1}{1/(A_1 - B_1 VPD) - 1}$$

The empirical factors $A_1$ (unitless) and $B_1$ (kPa$^{-1}$) are optimised in this work.

**Respiration**

$Q_{10}$ (unitless) used to calculate the temperature control of heterotrophic respiration:

$$cT = min(1, Q10^{((T-30)/10)})$$

where T is the surface/soil temperature for the above/below-ground pools.

The growth respiration is calculated as a fraction of the remaining total biomass:

$$Rg = FRAC_{growthresp} \cdot max (B_a - \delta t \cdot \sum R_{m,i} 0.2 \cdot B_a)$$

where $B_a$ is the total biomass, 1t the time step (one day), and $FRAC_{growthresp}$ a fraction to be optimised.

- Is the model calibration done starting always with the same spinup?
Yes. This has been clarified in the text:

> Once spun up, we performed two main sets of optimisation **always starting from this spinup**

- Revise terminology: You write in several places that "parameters showing sensitivity to outputs" (l. 128). However, the Morris sensitivity analysis quantifies the sensitivity of an output variable to a certain parameter (not vice versa), given that the parameter varies within the constrained range.
We agree that this is somehow misleading and we have revised the line as follows:

> we removed all parameters **to which** the different modelled outputs tested (i.e., net primary product (NPP) and leaf-area index (LAI)) **showed no sensitivity**. All remaining parameters were optimised in this study

- It is unclear what has changed in the model since Vuichard et al., 2019.
The version of ORCHIDEE-CN (r4999) described and evaluated in Vuichard et al. (2019) was based on a version of ORCHIDEE without the nitrogen cycle (trunk version r3977) which was anterior (by ~one year) to the one used for the CMIP6 exercise (ORCHIDEE_2.0, trunk version r5107). The version of ORCHIDEE_3 (branch ORCHIDEE_3 r6863) used in this study is based on ORCHIDEE-CN (r4999) but has been updated for latest developments of ORCHIDEE_2.0 regarding small ug corrections. In addition, few specific N-related process modelling has been

updated in ORCIDEE_3 (r6863) in particular growth and maintenance respiration modelling. The following has been added to L121:

> The version of ORCHIDEE used in our study (ORCHIDEEv3, r6863) is more recent than the one used Vuichard et al. (2019**, r4999**). **ORCHIDEEv3 (r6863) includes the latest developments of the main ORCHIDEE model (mainly small bug fixes). Furthermore, it includes updates to a few specific N-related processes, notably growth and maintenance respiration. Although this version** has been used in the multi-model ensemble for the Global Carbon Budget 2020 (Friedlingstein et al., 2022), **it** has not yet been optimised against independent data. As such, the initial fit of the model to the Fluxnet data is different than that shown in Vuichard et al. (2019).

**Interpretation of the calibration setup in the context of model improvement, applicability, and generalisability:**

- You write that "By optimising the model against a number of different constraints, we gain confidence in our parameter and hence in the projections". Note that a parameter optimisation, by design, always improves the model-observation fit, yet the model could be overfitted and perform poor on data that was not used for calibration. Hence, interpreting the value of this exercise as a means to "gain in confidence" is not generally justified. Indeed, the optimisation deteriorated some aspects of the fit, as documented here. Therefore, the discussion should address the generalisability of the calibrated model and potential of overfitting, in particular in view of site/experiment peculiarities (see comments by Martin DeKauwe).

We agree that this sentence can be a bit misleading. The point we were trying to make was that by using different constraints (data streams) we decrease the risk of overfitting and thus increase the chance of gaining confidence in the projections. Indeed, there will always be a risk of overfitting. Our study provides a way to expand the assimilation of classic observations (Fluxnet, Satellite phenology, temperature, soil moisture,...) with some observations under different $CO_2$ levels, which should hopefully decrease the risk of overfitting. We have expanded the text in the following manner:

> "However, we do need to be cautious in assessing these results since we are only using one FACE site for each PFT meaning we are likely tuning to the specificities of that site. For example, ORNL shows a progressive nitrogen limitation but this is not expected over all sites. Ideally, we would include a lot more FACE sites to capture different conditions. Especially, if we could optimise by grouping sites based on different levels of nitrogen limitation, then if the posterior parameters were found to be similar then the model processes allow for these differences.
>
> **In any optimisation, there is always a danger of overfitting to data limiting the generalisability of the calibrated model.** By optimising the model against a number of different constraints **(i.e. more than one data stream)**, we **decrease the risk of overfitting and therefore,** gain **some** confidence in our parameters and hence in the projections."

**Interpretation of the main results:**

- The finding "strength of the CO2 fertilization effect changes depending on the type of forest considered" is difficult to understand. What makes the two forest types investigated here (TeNE and TeBS) different in their C-N coupling and response to CO2? To what extent are specific information about experimental sites and setups underlying your results? Are different simulated responses to eCO2 at the two sites due to the parameter values or initial conditions? If ORNL parameters were used for the Duke simulations, would we also get a declining response ratio?

The $CO_2$ fertilisation is slightly different after the optimisation between the sites. This is probably due to the differences in site history, especially the soil fertility. Furthermore, the types of forests have very different functions (broadleaf vs needleleaf) most likely impacting their CO2 sensitivity. This difference in functions (both for the leaf type (broadleaf and needleleaf) and the leaf seasonality (summergreen vs evergreen)) means that testing the ORNL parameters at the DUKE site won't make much sense.However, since it is hard to disentangle the different responses here, we have decided to drop the sentence to avoid confusion.

- I feel like there is a somewhat selective reporting of results that leads readers to conclude that previously unconsidered FACE observations imply a smaller response of GPP to CO2 than when only FLUXNET data is used (You write: "we find that the rate of CO2 fertilisation is much lower when Free Air CO2 Enrichment data has been in the optimisation."). However, the response of GPP to CO2 that considered FACE observations in the calibration yields the same response as the model with prior parameter values (which excluded FACE data from the calibration). Please adjust the text to reflect this aspect.

We apologise for this confusing statement. This was alluding to the fact that the Fluxnet optimisation (blue) and the Fluxnet+FACE optimisation (orange) gave different GPP responses in the Figure 6. The prior model was not included in this statement, since this new version of the ORCHIDEE model had never been formally (Bayesian approach) calibrated. Indeed, the prior values changed a lot during our study period based on different manual tuning experiments. Nevertheless, we agree that the statement is too strong and can be misleading, especially in the abstract. Therefore the sentence has expanded as follows:

> "Using an idealised simulation experiment based on increasing atmospheric CO2 by 1% per year over 100 years, we find that **optimising against only FluxNet data tends to imply a large fertilisation effect whereas optimising against FluxNet and FACE data (with all nutrients limitation and acclimation of plant) decrease it significantly**."

- Negative response ratio of NPP simulated for Duke: The explanation provided doesn't seem to fit. An increasing autotrophic respiration (decreasing biomass production efficiency) should be expected if leaf N increases, not decreases.

We have changed autotrophic respiration to maintenance respiration in the text to be more correct, since, in the model, maintenance respiration is a function of leaf N but not growth respiration.

**Furthermore, the manuscript needs careful editing to resolve, among others, the following points:**

- Make sure that the use of tenses is consistent throughout the manuscript.
Done

- Since you don't have a Discussion section and since "discussion points" are wrapped within the Results section, I suggest to re-name Section 3 as "3. Results and Discussion".
Done

- l. 54 "One such experiment, the Free Air CO2 Enrichment experiment" - 'FACE' refers to an experimental setup and technique, not a particular experiment per se. There have been a number of FACE experiments conducted.
Changed to:
> One such **type of manipulation** experiment, the Free Air CO$_2$ Enrichment experiment**s** provide

- l. 55: "FACE experiments are conducted across several 55 vegetation types and typically consist of two plots" - Usually, there are more then one plot under each, the treatment and control.
Changed to:
> FACE experiments are conducted across several vegetation types and typically consist of two **types** of plots: one where CO is fumigated to high concentrations and one left as a control.

I am therefore returning the paper to you so that you can make the necessary (major) revisions and I am looking forward to receiving your revised manuscript.

Beni Stocker